Inflammatory bowel disease biomarkers of human gut microbiota selected via different feature selection methods

Bakir-Gungor Burcu burcu.gungor@agu.edu.tr 1
Hacılar Hilal 1
Jabeer Amhar 1
Nalbantoglu Ozkan Ufuk 2
Aran Oya 3
Yousef Malik 4 5
1 Department of Computer Engineering, Abdullah Gul University , Kayseri , Turkey
2 Department of Computer Engineering, Erciyes University , Kayseri , Turkey
3 TETAM, Bogazici University , Istanbul , Turkey
4 Zefat Academic College , Zefat , Israel
5 Galilee Digital Health Research Center, Zefat Academic College , Zefat , Israel
Setubal Joao
Electronic publication date: 2022 Apr 25
Publication date: 2022
Volume: 10
Electronic Location ID: e13205
Received 2021 Feb 3; Accepted 2022 Mar 10
Copyright: ©2022 Bakir-Gungor et al.
Copyright year: 2022
Copyright holder: Bakir-Gungor et al.
License: This is an open access article distributed under the terms of the Creative Commons Attribution License, which permits unrestricted use, distribution, reproduction and adaptation in any medium and for any purpose provided that it is properly attributed. For attribution, the original author(s), title, publication source (PeerJ) and either DOI or URL of the article must be cited.
License URL: https://creativecommons.org/licenses/by/4.0/

Keywords: Feature selection, Human gut microbiome, Biomarker discovery, Classification, Metagenomics

Funding: The Abdullah Gul University Support Foundation (AGUV) The Zefat Academic College The work of Burcu Bakir-Gungor has been supported by the Abdullah Gul University Support Foundation (AGUV). The work of Malik Yousef has been supported by the Zefat Academic College. The funders had no role in study design, data collection and analysis, decision to publish, or preparation of the manuscript.

==============================
The tremendous boost in next generation sequencing and in the “omics” technologies makes it possible to characterize the human gut microbiome—the collective genomes of the microbial community that reside in our gastrointestinal tract. Although some of these microorganisms are considered to be essential regulators of our immune system, the alteration of the complexity and eubiotic state of microbiota might promote autoimmune and inflammatory disorders such as diabetes, rheumatoid arthritis, Inflammatory bowel diseases (IBD), obesity, and carcinogenesis. IBD, comprising Crohn’s disease and ulcerative colitis, is a gut-related, multifactorial disease with an unknown etiology. IBD presents defects in the detection and control of the gut microbiota, associated with unbalanced immune reactions, genetic mutations that confer susceptibility to the disease, and complex environmental conditions such as westernized lifestyle. Although some existing studies attempt to unveil the composition and functional capacity of the gut microbiome in relation to IBD diseases, a comprehensive picture of the gut microbiome in IBD patients is far from being complete. Due to the complexity of metagenomic studies, the applications of the state-of-the-art machine learning techniques became popular to address a wide range of questions in the field of metagenomic data analysis. In this regard, using IBD associated metagenomics dataset, this study utilizes both supervised and unsupervised machine learning algorithms, (i) to generate a classification model that aids IBD diagnosis, (ii) to discover IBD-associated biomarkers, (iii) to discover subgroups of IBD patients using k-means and hierarchical clustering approaches. To deal with the high dimensionality of features, we applied robust feature selection algorithms such as Conditional Mutual Information Maximization (CMIM), Fast Correlation Based Filter (FCBF), min redundancy max relevance (mRMR), Select K Best (SKB), Information Gain (IG) and Extreme Gradient Boosting (XGBoost). In our experiments with 100-fold Monte Carlo cross-validation (MCCV), XGBoost, IG, and SKB methods showed a considerable effect in terms of minimizing the microbiota used for the diagnosis of IBD and thus reducing the cost and time. We observed that compared to Decision Tree, Support Vector Machine, Logitboost, Adaboost, and stacking ensemble classifiers, our Random Forest classifier resulted in better performance measures for the classification of IBD. Our findings revealed potential microbiome-mediated mechanisms of IBD and these findings might be useful for the development of microbiome-based diagnostics.

Introduction

Human gut microbiota is a complex community of microorganisms including trillions of bacteria that populate in our gastrointestinal tract. Although some of these microorganisms are considered as essential regulators of our immune system, the dysbiosis of the human-associated microbial communities has been linked with several diseases (Wang & Liu, 2020). The modulation of the complexity and eubiotic state of microbiota might lead to autoimmune and inflammatory disorders such as diabetes, obesity, rheumatoid arthritis, inflammatory bowel diseases (IBD), and carcinogenesis (Li et al., 2018; Lazar et al., 2018). In this respect, deciphering the function and composition of our gut microbiome—the collective genomes of the microbial community that reside in the human gut is—crucial (Halfvarson et al., 2017).

In recent decades, the rapid advances in next-generation sequencing (NGS) technologies enabled the generation of millions to billions of reads in a single run. Metagenomic NGS approaches, which allow the analysis of the entire genomic content of a sample and provide taxonomic and functional profiles of microbial communities, accelerated the discovery of the human gut microbiome. Since the gut microbiome is modulated via human-microbiome symbiosis, the metagenomic analysis of the gut microbiome provides novel insights regarding the effect of human gut microbiota on human physiology and diseases (Manichanh et al., 2012; Frank et al., 2007; Greenblum, Turnbaugh & Borenstein, 2012; Peterson et al., 2008; Mandal, Saha & Das, 2015; Ditzler, Polikar & Rosen, 2015).

IBD, comprising Crohns disease (CD) and ulcerative colitis (UC), is a gut-related disorder that affects the intestinal tract. The most widely reported symptoms of IBD involve diarrhea, intermittent nausea and vomiting, rectal bleeding, and abdominal pain or tenderness (Tavakoli et al., 2021; Baumgart & Sandborn, 2007; Strober et al., 2007). Most of these symptoms occur due to intestinal damage which happens as a result of the exaggerated inflammatory response. Although the development and progression mechanisms of IBD are poorly understood, multiple factors (involving genetic, physiological, immunological, psychological environmental factors, gut microbiome) and their interactions are considered to have a role (Tavakoli et al., 2021). One of the factors that are considered to promote the development of IBDs is an aberrant immune response against commensal bacteria (Lazar et al., 2018). Recently, various aspects of the immune system-microbiota crosstalk, such as microbial sensing, oxidative stress, and antigen processing are studied using different experimental models (Lazar et al., 2018). During IBD pathogenesis, firstly different types of bacteria bind to the gut mucosa and conquer into mucosal epithelial cells. Secondly, this event triggers an inflammatory response, which is interposed by the production of TNF- by monocytes/macrophages. Thirdly, this chronic bowel inflammation influences the tolerance of the epithelial cells to intestinal bacteria. Finally, this situation causes alterations in the intestinal microbiota composition such as an elevation in aerobic bacteria, which causes an important reduction in the fecal levels of propionic acid and butyric in IBD patients (Lazar et al., 2018).

There is high comorbidity of IBD with other dysbiosis-related illnesses, such as depression, anxiety, or obesity. A recent review paper (Scotti et al., 2017) outlined the tight connection between the gut, microbiota, and brain in the context of IBD. The authors have especially focused on the effect of stress on this interplay (Gao et al., 2018; Oligschlaeger et al., 2019). Gut microbiota can also produce neurochemicals having hormonal activities. These neurochemicals could reach further than the gut and they have roles in the modulation of depression, anxiety, pain, cognition, autoimmune, inflammatory and metabolic diseases (Lazar et al., 2018; Grenham et al., 2011; Haemer, Huang & Daniels, 2009; Bercik et al., 2011; Ochoa-Repáraz et al., 2011; Erny et al., 2015). Compared to the expected prevalence of both depression and anxiety in the general population, the prevalence estimate of these diseases was higher in IBD patients (Baxter et al., 2013; Ferrari et al., 2013). The relationships between symptoms of depression and anxiety with more severe IBD symptoms, more episodes of relapse in IBD patients (Mittermaier et al., 2004), and higher rates of hospitalization were noted in the literature (Van Langenberg et al., 2010). In this respect, reported co-morbidities of depression and high anxiety and depression in IBD patients have suggested a causal relationship between IBD symptoms and these two conditions (Km et al., 2007; Tavakoli et al., 2021). IBD is also considered to influence the development of obesity (Harper & Zisman, 2016; Flores et al., 2015). Important variations in the composition of the intestinal microflora are reported in people having excess body weight (Jarmakiewicz-Czaja, Sokal & Filip, 2020). Implementation of proper dietary ingredients that generate desirable effects on the gut microbiota is very important for both obesity and IBD patients (Chen et al., 2020). In other words, preserving healthy intestinal microflora is the key to reduce the body mass index (BMI) and to alleviate the course of IBD.

As presented above, the deviations from the “healthy” gut microbiome that occur due to environmental effects, dietary and genetic mutations are one of the factors that are thought to be associated with IBD. Hence, the metagenomic analysis of the human gut microbiome helps to illuminate microbiome-associated factors of IBD. To detect gut-related diseases, researchers believe that for some cases the diagnostic use of the microbiome is possible before detecting via conventional diagnostics (Marchesi et al., 2016). IBD is one of such gut-related diseases where the aetiology is not completely understood and the symptoms are complex. In this respect, the design of new tools that make use of the available human gut metagenome data is essential for the diagnosis of IBD. To this end, machine learning (ML) is well suited to obtain a diagnostic model using an IBD-associated metagenomics dataset (Tabib et al., 2020).

Recently several studies attempted to use ML for the analysis of large metagenomics datasets (Mandal, Saha & Das, 2015; Pasolli et al., 2016; Nguyen et al., 2018; LaPierre et al., 2019). A general overview of ML approaches for metagenomics studies has been provided in different reviews (Soueidan & Nikolski, 2015; LaPierre et al., 2019). Pasolli et al. (2016) worked on the classification of the patients and the controls (healthy samples) using the metagenomic-associated datasets of colorectal cancer, cirrhosis, obesity, IBD, and type II diabetes (T2D). They tested the performances of the Support Vector Machines (SVM), Random Forest (RF) classifiers and also evaluated Lasso (Tibshirani, 1996) and Elastic Net (Zou & Hastie, 2005) regularized multiple logistic regression. They processed the datasets of different diseases using the same bioinformatics pipeline (Metagenomic prediction Analysis based on Machine Learning, MetAML) (Pasolli et al., 2016). Due to the low number of CD patients, they combined CD and UC patients and their predictive model achieved an Area Under the Curve (AUC) of 0.914 for discrimination of IBD patients using the species abundance information as a feature vector.

To diagnose an individual’s disease status based on microbiome module abundance, Nguyen et al. (2018) used a random forest classifier. While they could achieve an AUC of 0.954 for CD, they obtained an AUC of 0.783 for UC. Via experimenting on the microbiome data of some other diseases such as T2D, colorectal cancer, liver cirrhosis, obesity and rheumatoid arthritis, they concluded that the classification sensitivity and specificity metrics of classification vary as a function of disease.

To improve the classification of diseases with metagenomic data, Armour et al. (2019) proposed to represent metagenomic data as images and used Convolutional Neural Networks (CNNs) for classification. Their procedure named Met2Img achieved an accuracy of 0.842 (Armour et al., 2019). Reiman, Metwally & Dai (2018) implemented a framework called PopPhy-CNN. To depict the relatedness of various features, i.e. microorganisms, PopPhy-CNN utilizes phylogenetic trees. The tree is further embedded in a 2D matrix. Through this 2D matrix, one can exploit the relative abundances and relationships of microbial taxa and their quantitative characteristics in metagenomic data. Reiman, Metwally & Dai (2018) have shown that their framework can effectively train models without an excessive amount of data.

For metagenome-based disease prediction, LaPierre et al. (2019) evaluated different methods that apply deep learning and machine learning. They compared MetAML, which uses RF and SVM (Pasolli et al., 2016); PopPhy-CNN (Reiman, Metwally & Dai, 2018) and Met2Img (Nguyen et al., 2018), which are CNN based methods; and RegMIL (Rahman & Rangwala, 2018), which models the problem with Multiple Instance Learning (MIL), while using both a neural network and RF as part of its pipeline. LaPierre et al. (2019) noted that, as the performance evaluation metric, PopPhy-CNN (Reiman, Metwally & Dai, 2018) reports AUC, Met2Img (Nguyen et al., 2018) reports accuracy, RegMIL (Rahman & Rangwala, 2018) reports both AUC and accuracy, and MetAML (Pasolli et al., 2016) reports AUC, accuracy, recall, precision, and F1 score. In their comparative evaluation, while Met2Img-CNN resulted in the best accuracy value of 0.868, MetAML-RF resulted in the best AUC of 0.890 for the IBD metagenomics dataset (LaPierre et al., 2019). Since Pasolli et al. (2016), Armour et al. (2019), Nguyen et al. (2018) and LaPierre et al. (2019) did not apply any feature selection method, they require the use of abundance information for all species in their classification model.

 Wingfield et al. (2016) presented a method for the stratification of IBD presence, and the identification of IBD subtype from a bacterial census of the intestinal microbiome. Using a hybrid classifier of Multi-Layer Perceptron (MLP) and SVM, they obtained AUCs of 0.70 and 0.74 for CD and UC, respectively. For the diagnosis of IBD, Hacılar, Nalbantoğlu & Bakir-Güngör (2018); Hacilar et al. (2020) analyzed IBD-associated metagenomics dataset using different machine learning algorithms, ensemble methods, and shrinkage methods including ridge regression, Lasso and Elastic Net. In experiments with 10-fold cross-validation, the best models achieved an AUC of 0.919, an accuracy of 87.7 % and a F1-measure of 83.7 %. In this study, firstly, we attempt to identify IBD-associated biomarkers via utilizing robust feature selection algorithms such as Conditional Mutual Information Maximization (CMIM) (Brown et al., 2012), Fast Correlation Based Filter (FCBF) (Fleuret, 2004), Min redundancy Max relevance (mRMR) (Ding & Peng, 2005), select K best (SKB) (Pedregosa et al., 2011), Information Gain (IG) and Extreme Gradient Boosting (XGBoost) (Ditzler et al., 2015). Using 100-fold Monte Carlo Cross-Validation, state-of-the-art machine learning algorithms, and ensemble classification methods (i.e. SVM, Decision tree, RF, Adaboost, and Logitboost), the performance of the reduced metagenomics dataset has been assessed. The models have been evaluated systematically and extensively using several performance metrics. Secondly, this study aims to find subgroups of IBD patients via applying K-means and hierarchical clustering on the IBD-associated metagenomics dataset. Since the symptoms and the treatments of IBD are complex, the precise detection of IBD subgroups could provide valuable insights for identifying individualized therapy targets and will pave the way towards personalized medicine applications. Additionally, Principal Component Analysis (PCA) is performed to obtain the underlying structure of IBD metagenomics data. In summary, this study utilizes both supervised and unsupervised machine learning methods to (i) build a classification model that facilitates IBD diagnosis, (ii) find out potential pathobionts of IBD, and (iii) discover subgroups of IBD patients.

The rest of this paper is organized as follows: Section 2 presents the machine learning algorithms and feature selection methods that are used to obtain a diagnostic model of IBD, to identify IBD biomarkers of human gut microbiota, and to discover the subgroups of IBD patients. Section 3 highlights the findings and provides an extensive evaluation of the presented method. Section 4 is devoted to the discussion of the findings. Section 5 concludes the paper and summarizes avenues for further research.

Materials and Methods

In disease prediction problem, mainly three types of features could be generated from metagenomic sequence reads: (i) the abundances of different microbes, (ii) functional annotation of the metagenomic samples, (iii) the k-mer abundances obtained from raw reads. The identification and the quantification of different organisms is one of the main tasks in metagenomic sequence data analysis. Since the microbiome composition is different between cases and controls, the microbial abundance profiles are commonly used as a feature in disease prediction (type (i) of the above-mentioned features).

Dataset and preprocessing steps

This study aims to develop a classification model to aid IBD diagnosis and to discover IBD-associated bio-markers using metagenomics data. In this respect, the raw microbiome DNA sequencing data of 148 IBD patients and 234 control samples were fetched from the MetaHit project (Oudah & Henschel, 2018; ERA000116). Prior to taxonomic analysis, we followed the standard quality analysis, which is proposed by the Human Microbiome Project (HMP) (Consortium et al., 2012; Young, 2011). According to that, firstly the duplicate reads were removed by a modified version of EstimateLibraryComplexity in Picard tool package. The quality trimming was performed using TrimBWAStyle script (Fass, 2010), which is also recommended by the HMP SOP. The human genome contamination removal step was skipped, as the provided data is already free of human contamination. Prior to taxonomic analysis, the reads smaller than 90 bp in length were filtered out.

To estimate the relative abundance of microbial taxa, we use the MetaPhlAn2 tool (Ditzler, Polikar & Rosen, 2015), which is a widely used method in the literature. MetaPhlAn2 firstly assigns reads to microbial clades using a set of clade-specific marker genes; and secondly, it estimates the relative abundance of microbial taxa based on the read coverage. In this study, each DNA sequence is assigned to its microbial species of origin (taxa) using MetaPhlAn2 taxonomic classification tool. The standard relative abundance normalization procedure is followed by dividing the read count of each taxonomic bin by the total number of the reads for a sample. Hence taxonomic abundances values are obtained as real numbers in the range of [0, 1], which sums up to 1 within each sample. Samples containing less than one million reads were discarded. Consequently, the microbial diversity (i.e. which microorganisms exist in what relative abundance) of the gut microbiome for each sample was revealed.

Microbiome sequencing data is categorized into disease states based on the associated metadata. In the original study that provides the metagenomics data (PRJEB2054; PRJEB2054), the healthy controls were selected among the close relatives of IBD patients and the individuals who had antibiotic treatment for at least 4 weeks before fecal sample collection was not included. IBD subjects were in clinical remission for at least 3 months, and had stable maintenance therapy with azathioprine or mesalazine. The patient demographics providing the country, gender, age, body-mass index are available in the original study (PRJEB2054).

At the end of the data preprocessing steps, the dataset included the relative abundance values for 3,302 different taxa for 382 samples. In the lower taxonomic levels (species), there are considerable variations in human gut microbial composition (Eckburg et al., 2005). Hence, species-level information is widely used in metagenomics classification problems. In this respect, the features of kingdom, phylum, class, order, family, genus were removed, and the remaining 1,455 features were considered. When the features of the same species for different strains were combined into one single feature, the final dataset included 1,331 species-level features. As shown in Fig. 1, the final IBD-associated metagenomics dataset is composed of the relative abundance values for 1,331 different species for 148 IBD patients and 234 healthy samples. This dataset is used to develop, train, test, and validate a machine learning model to diagnose IBD.

Figure 1 Illustration of the inflammatory bowel disease-associated metagenomics dataset.

The methodology of this study can be summarized by the following steps: (i) feature selection to identify the most informative IBD-associated biomarkers; (ii) model construction to classify IBD patients and control samples, followed by the assessment of the generated models using extensive evaluation metrics; (iii) unsupervised learning to detect subgroups of IBD patients and control samples. Figure 2 shows the details of these three steps.

Figure 2 Schematic representation of the methodology.

(i) Feature selection methods (shown in red) are applied to detect the most important species for the development of IBD (IBD-associated microorganisms), (ii) Using the selected features, models are constructed and used for classification (shown in blue), (iii) K-means clustering algorithm is applied on data to discover subgroups of IBD patients and control samples (shown in green).

Feature selection

Feature selection is known as an effective preprocessing tool for machine learning problems (Tang, Alelyani & Liu, 2014). However, deciding between the growing numbers of feature selection methods is challenging. Different feature selection methods have their advantages/disadvantages as discussed in different reviews (Chandrashekar & Sahin, 2014; Khalid, Khalil & Nasreen, 2014; Li, Li & Liu, 2017). In a previous work (Manikandan & Abirami, 2021), several state-of-the-art feature selection methods were reviewed in terms of their ability to overcome common problems such as data nonlinearity, correlation and redundancy, noise in the target class, noise in the input features, and having a much higher number of features much higher than the number of samples. Feature selection methods have different usages in the biomedical domain, as presented in Remeseiro & Bolon-Canedo (2019). Moreover, there are different kinds of feature selections, i.e. filter, wrapper, and embedded methods such as (Yousef et al., 2020). Recent feature selection methods make use of the biological knowledge, which is embedded in the machine learning algorithm (Yousef, Sayıcı& Bakir-Gungor, 2021; Yousef, Abdallah & Allmer, 2019; Yousef et al., 2021). Applications of biological domain knowledge based feature selection methods for gene expression data can be found in: Yousef, Kumar & Bakir-Gungor (2021).

In metagenomics studies, the number of predictors (number of taxa or features) is much more than the number of observations (samples). This phenomenon is known as the curse of dimensionality. In this respect, some metagenomics studies focus on the feature selection process rather than focusing on classification. It has been discussed in the literature that although the feature selection process in metagenome-based disease prediction problem is relatively less studied, this area could be as critical as the choice of the classification method (LaPierre et al., 2019). The feature selection process in metagenome-based disease prediction problem could improve the elucidation of the disease mechanisms. Hence, further research in this direction is provoked. To reduce the number of taxa (species or features), in other words, to select informative features, min Redundancy Max Relevance (mRMR) (Ding & Peng, 2005), Lasso (Tibshirani, 1996), Elastic Net (Zou & Hastie, 2005) and iterative sure select algorithm (Duvallet et al., 2017) have been extensively applied previously. Another feature selection method called Fizzy challenges to detect important microbes or functional elements for downstream analysis using classification algorithms (Ditzler et al., 2015). Oudah and Henschel introduced a competing taxonomy-aware feature selection method (Oudah & Henschel, 2018). (Bakir-Gungor et al., 2021) applied CMIM (Brown et al., 2012), FCBF (Fleuret, 2004), mRMR (Ding & Peng, 2005), and Select K best (SKB) (Pedregosa et al., 2011) on Type 2 diabetes associated metagenomics dataset and obtained high-performance metrics. Although those feature selection approaches are well studied in different domains, they are just getting attention in this domain. In a recent review paper (Marcos-Zambrano et al., 2021), some of these methods are reported to achieve good results in human microbiome studies. However, as noted by this review paper, for metagenomics studies, there is no consensus on which feature selection method should be used. In this study, we proposed that multiple feature selection methods should be used and the final features are determined by getting the intersection of the features selected by different methods. Conditional Mutual Information Maximization (CMIM) (Brown et al., 2012), Fast Correlation Based Filter (FCBF) (Fleuret, 2004), Min Redundancy Max Relevance (mRMR) (Ding & Peng, 2005), SKB, Information Gain (IG) and Extreme Gradient Boosting (XGBoost) (Chen & Guestrin, 2016) feature selection algorithms are applied on the metagenomics dataset.

Information Gain (IG) is one of the filter-based feature selection techniques, which leverages the concept of entropy (Gray, 2011; Kent, 1983). More specifically, a weight for each feature is calculated by analyzing the reduction in the class entropy (to which extent the uncertainty in the class prediction drops) when the value of that feature is known. IG is one of the widely used feature selection techniques, and it has applications in different domains (Bolón-Canedo & Alonso-Betanzos, 2019).

mRMR is another mutual information (MI) based method, and it chooses features according to the maximal statistical dependency criteria. Since the implementation of the maximal dependency condition is laborious, mRmR works as an approximation technique to maximize the dependency between the joint distribution of the selected features and the classification variable. In other words, mRMR (Ding & Peng, 2005) attempts to choose the features that have the minimum correlation between themselves (defined as min redundancy); and the maximum correlation with a class to predict (known as max relevance).

CMIM (Brown et al., 2012) first ranks the features according to their conditional entropy and MI with the class to predict; and then selects the feature if it carries additional information. Likewise, FCBF (Fleuret, 2004) ranks the features based on their MI with the class to predict; and then removes the features whose MI is less than a predefined threshold. This method aims to detect relevant features, as well as redundancy among relevant features via calculating feature-class and feature-feature correlations. FCBF picks features in a classifier-independent manner. It chooses the features with high correlation with the target variable, but low correlation with other variables. As a correlation measure, FCBF uses Symmetrical Uncertainty, which harmonizes Shannon entropy and IG. Select K best method scores the features against the class label using a function, and it selects the features having k highest score (Pedregosa et al., 2011). In XGBoost (Chen & Guestrin, 2016) feature selection, the more an attribute is used to make key decisions with decision trees, the higher relative importance it gets. In this study, CMIM, mRMR, FCBF, SKB, IG and XGBoost feature selection methods are implemented in Python 3, using the skfeature and sklearn libraries.

Model construction

Applying supervised learning to the human gut microbiome is useful to detect subsets of microorganisms that are substantially discriminative. Accordingly, one can generate prediction models that can precisely classify unlabeled samples. Recently, Marcos-Zambrano et al. (2021) reviewed machine learning (ML) applications for microbiome studies via analyzing 89 papers. They reported that the most common supervised learning algorithms that were used for microbiome analysis were Random Forest (RF), Support Vector Machines (SVM), Logistic Regression (LR) and k-NN (k nearest neighbor). They concluded that multiple factors need to be taken into account while selecting the ML algorithm (i.e., number of observations, number of features, data type, data quality etc.), and they recommend applying and evaluating more than one method and selecting the one with the highest performance value. Another recent study by Wang & Liu (2020) compared the performances of two ensemble methods (RF, eXtreme Gradient Boosting decision trees (XGBoost)), and two traditional methods (Elastic net and SVM) on 29 human microbiome benchmark datasets. They find that in a few benchmark datasets XGBoost outperforms all other methods; and in the remaining benchmark datasets, XGBoost, RF and Elastic net were comparable. Along this line, in this study, a range of machine learning models were constructed to discriminate IBD samples from controls, using different classification algorithms (RF, Decision Tree, Logitboost, Adaboost, SVM, and stacking ensemble classifiers (i.e., an ensemble of SVM with kNN, an ensemble of the Logitboost with kNN). The working principles of these classification algorithms can be summarized as follows:

Decision trees are the base learners of RF. Each tree is a non-linear model and each model is created with many linear boundaries (Liaw & Wiener, 2002). During the splitting (training) procedure, RF randomly chooses bootstrapped samples from the original data, and randomly selected subsets of features are used to assess the quality of the model. Finally, RF merges several decision trees into a single ensemble model and performs prediction via combining the predictions of individual trees.

In kNN classification, for each sample, the class labels of k training samples nearest to that point (neighbor samples) are investigated, and the majority of those class labels are assigned to the sample. To specify the neighborhood, the most widely used distance metrics are correlation coefficients and Euclidean distance.

SVMs attempt to find out a decision boundary between the classes. This decision boundary helps to assure the maximum feasible distance or margin between the samples that are closest to this decision boundary. The decision boundary is defined via learning from the samples that are closest to the boundary, and hence these samples are called support vectors. In some cases, a linear separation between the classes is not feasible in the original feature space. In such cases, SVM utilizes the kernel trick to determine the decision boundary in a higher-dimensional space (Cortes, 1995).

Boosting was initially proposed (Schapire, 1990) to unite multiple weak classifiers and hence to elevate the classification performance. Freund & Schapire (1997) proposed a more practical and capable boosting algorithm called AdaBoost. AdaBoost, which stands for Adaptive Boosting, is one of the most widely used algorithms with many applications in numerous fields. AdaBoost focuses on the difficult samples by assigning higher weights on them that could not be properly classified with the previous weak classifiers. AdaBoost is capable of reducing the training errors exponentially fast as long as the weak classifiers produce just better results than the random case (Freund & Schapire, 1997). It was reported that AdaBoost had very good generalization capability. Still, like most other classifiers, when dealing with very noisy data, AdaBoost suffers from the overfitting problem (Rätsch, Onoda & Müller, 2001). To overcome this situation, Schapire, Freund et al. (1995) proposed another algorithm called LogitBoost, which could reduce training errors linearly, and hence better generalization. It was designed to decrease the bias and the variance. Logitboost was originally utilized for integrating non-complex classifiers to improve performance in classification. This technique optimizes the multinomial likelihood, which makes it easy to apply in multiclass problems. The derivation of the LogitBoost algorithm can be done by applying logistic regression to the AdaBoost generalized additive model.

Throughout the experiments of this study, 100-fold MCCV is used. In MCCV, some part of the data is randomly chosen (without replacement) as the training set and the rest is used as the test set (Xu & Liang, 2001). This procedure is iterated several times, and hence new training and test sets are randomly generated in each iteration. 90 % is chosen for training and 10 % is chosen for testing. As shown in the workflow Fig. 2, the feature selection methods are applied on the training set. Classification methods are tested on the testing part. The Konstanz Information Miner (KNIME) platform (Berthold et al., 2009) is used for the implementation of the methodology. H20 library in KNIME and Python scikit-learn library (Pedregosa et al., 2011) are also used.

Model performance evaluation

Prediction performances of the generated models were evaluated by using accuracy, F1 Score, and AUC measures. Accuracy is a widely used performance evaluation metric and a reliable measure for balanced datasets. Since there is an uneven class distribution in the dataset of this study, other metrics such as the F1 score and AUC have been utilized to evaluate the performance of the generated models. Precision is defined as the percentage of the predicted cases that are actual cases. On the other hand, recall represents the percentage of actual cases that are correctly identified by the classifier. That is to say that recall determines the rate of falsely predicting healthy. Precision denotes the rate of falsely predicting disease. F1 score is calculated as the harmonic mean of precision and recall. Among different performance metrics, F1 score is a good choice when someone seeks a balance between precision and recall, and when there is an uneven class distribution (a large number of actual negatives). Several classifiers deliver the probability values for their predictions, which can be considered as their confidence values regarding the prediction. The AUC utilizes this information to recapitulate the false prediction rate at different confidence levels. AUC is commonly used as a summary measure of diagnostic accuracy. In real-life examples, there is an overlap between the test results of positive and negative examples. AUC shows how the recall vs. precision relationship changes as the threshold or the cut-off value for identifying a positive is changed. All performance results presented in this study refer to the average of 100-fold MCCV.

Unsupervised learning

One of the goals defined for precision medicine is stratifying patients based on disease subtypes, disease progression, and response to treatments via analyzing molecular profiling data of the patients (Korcsmaros, Schneider & Superti-Furga, 2017). Precision medicine provides big promise to enhance the course of care for IBD patients since it offers the most effective therapy while minimizing the side effects (Weersma et al., 2018). In this respect, studying the relationships between the samples and the relative abundance values of the species can help to detect subgroups of IBD patients; and hence enlighten IBD development mechanisms. To analyze and visualize these relationships, in this study, three unsupervised learning approaches (K-means clustering, principal component analysis (PCA), and hierarchical clustering) were used for the following purposes. Firstly, this study attempts to answer whether there could be any direct relationship between specific species (or a group of species) and IBD subgroups. In this respect, to explore the subgroups of IBD patients and subgroups of healthy samples, K-means clustering is used. K-means (Steinley & Brusco, 2007) is an unsupervised learning algorithm that groups samples by minimizing the distance inside the clusters and maximizing the distance between the clusters. The Euclidean distance metric and Elbow method (Bonaros, 2019) are utilized in this study to find out the optimum number of clusters. In this method, the point where the decline in the error slows down indicates the optimum number of clusters. t-distributed Stochastic Neighbor Embedding (t-SNE) is a nonlinear dimension reduction technique. It is a commonly used graphical approach to guide clustering methods such as K-means in terms of deciding the number of clusters and cluster memberships. In this study, t-SNE is employed for visualizing the identified clusters (subgroups of IBD patients and healthy samples).

Secondly, to observe whether the relative abundance values of the species can induce the formation of two clusters representing IBD patients and controls, IBD-associated metagenomics data were modeled using principal component analysis (PCA). PCA is a dimensionality reduction algorithm that converts a high dimensional space (where each dimension corresponds to a species) to a lower-dimensional space (usually 2D or 3D). In unsupervised machine learning, the class labels (here the diagnosis of IBD, in other words cases and controls) are hidden from the model, leaving the algorithm to impose the most relevant strata.

Thirdly, to better visualize the relationship between the relative abundance values of the selected species and IBD patients/controls, hierarchical clustering was performed using Euclidean distance and Ward variance minimization algorithm as the linkage method. With this analysis, the aim is to reveal the presence of distinct subgroups of IBD patients, corresponding to the ones having complex patterns of microbial species.

Results

Feature selection and classification

Since the dysbiosis between the mucosal immune system and gut microbiota is a hallmark of IBD pathogenesis, the microbiome has the potential to harbor species with biomarker capacity. To detect those species with biomarker capacity, an IBD-associated metagenomics dataset, which includes the relative abundance values for 1,331 different species for 382 samples (following the preprocessing steps in Methodology section), is analyzed. This research effort attempted to remove the irrelevant and redundant features (species) using six different feature selection strategies (FCBF, CMIM, mRMR, SKB, IG and XGBoost). For each feature selection method, the top 100 features are investigated. To evaluate the effects of different classification methods, Decision Tree, Random Forest, LogitBoost, AdaBoost, an ensemble of SVM with kNN, and an ensemble of the Logitboost with kNN are considered. The parameters of c and gamma are optimized for SVM; number of trees is optimized for Random Forest, and n estimators is optimized for Logitboost and Adaboost. By using several metrics as described in the Methods section, the performances of different classifiers are compared using (i) all features (without feature selection); (ii) top 100 features selected using CMIM, mRMR, FCBF, SKB, IG and XGBoost (as presented in Table S1). As shown in the same table, SKB, IG and XGBoost feature selection methods resulted in high accuracy, F1 score, sensitivity, recall values, and high AUC scores for different classifiers throughout the experiments with IBD-associated metagenomics data. It can be observed from the same table that CMIM, FCBF, and MRMR feature selection methods showed signs of poor fitting across all models with low accuracy and high recall. To find the most relevant and informative features, for each feature, the scaled importance values were calculated for three selected feature selection methods (SKB, IG and XGBoost) across different classifiers (presented in Table S2). To eliminate the lowest ranking features among the top 100, the scaled importance value cutoff of 0.5 is applied. As shown in Fig. 3, this procedure generated 23, 57, 96 selected features in SKB, IG, and XGBoost feature selection methods, respectively. 14 of those features were commonly identified in all three-feature selection methods.

Figure 3 Numbers of selected species using different feature selection algorithms and the numbers of intersecting species among different feature selection methods.

As shown in Fig. 4 and Table S1, throughout the experiments using 14 features, it is observed that RF, LogitBoost, and Adaboost classifiers resulted in the top three performance results as compared with other classifiers. Among these three classifiers, using the 14 selected features, the RF classifier generated higher performance results for the above-mentioned promising feature selection methods (SKB, IG, and XGBoost). In RF, the interpretation of the tree model is simple and the model can be easily transformed into a ruleset. Also, as noted in Marcos-Zambrano et al. (2021)s comprehensive review, RF is one of the widely used algorithms in human microbiome studies. As shown in Table S1, even if all 1,331 features are used with Logitboost or with Adaboost, RF with 14 features performs as well as Logitboost and Adaboost, even slightly better. It should be noted that one of the goals of this study is to identify IBD-associated biomarkers of human gut microbiota. Therefore, obtaining satisfactory performance metrics for diagnosing IBD using a small number of selected species is important. The experiments conducted in this study showed that, on Exploration Cohort, the generated RF model using only 14 features resulted in adequate diagnostic accuracy (as shown in Fig. 4 and Table S1). For all these reasons, throughout the rest of the paper, we focused on the results obtained using the RF classifier.

Figure 4 Performance evaluations of different classifiers on IBD metagenomics dataset, utilizing 100-fold Monte Carlo cross-validation and using (A) XGBoost, (B) Select K Best, and (C) Information Gain feature selection methods, (D) 14 selected features, (E) all features.

When all 1,331 features are used (without applying feature selection), the generated RF model yielded 0.85 F1 score, 0.92 AUC, and 87 % accuracy on Exploration Cohort, as shown in Fig. 5 and Table S1. By only using the 14 features that are commonly identified in three promising feature selection methods, 0.85 F1-score, 0.93 AUC, and 88% accuracy metrics were obtained using the RF model. Compared with the performance values of the RF model using all features, the model with those selected 14 features performed 1 % higher in terms of accuracy and AUC metrics; 5 % higher in terms of specificity and precision metrics, as shown in Fig. 5 and Table S1. Also, using only those 14 species, the generated RF model yielded the same F1 score (0.85) as the one obtained using all features.

Figure 5 Comparative evaluation of different feature selection methods based on (A) Accuracy, (B) Area under ROC, and (C) F-Measure, using the Exploration Cohort dataset.

Validation on external data

The performance of the proposed method is also evaluated using an external dataset. A multicenter gut metagenome dataset (Project accession: PRJEB1220) with samples collected from Denmark and Spain, containing Ulcerative colitis (UC) and Crohns disease (CD) cases as well as healthy controls were considered for validation. A random subset of samples of 50 UC patients, 50 CD patients (in total 100 IBD patients) and 100 healthy controls, containing more than 1 Gbp of sequencing reads were subject to taxonomic analysis. MetaPhlAn 3.0 is run using default parameters and the relative abundances of all detected taxonomies are determined. The same preprocessing protocol that is used for the Exploration Cohort (accession: PRJEB2054), and that is presented in the Methods section; is followed for the independent test data (Project accession: PRJEB1220). While the Exploration data included 1331 species, the independent test data included 488 species as features. For the validation data, it is ensured that the species or genus names match, and the same reference database is used. Among the 14 potential taxonomic biomarkers, 10 species (Table S3) are found in the validation dataset. As shown in Table 1, throughout the experiments on the validation data using the selected species as features, it is observed that RF, LogitBoost, and Adaboost classifiers resulted in the top three performance results as compared with other classifiers. Among these three classifiers, RF performed slightly better (0.86 accuracy) than the other two classifiers (0.84, 0.82 accuracy). It is worth noting that the same trend was also observed for the Exploration Cohort, as shown in Fig. 4 and Table S1. Additionally, using 100-fold MCCV and the same set of classifiers, which are utilized in the experimentation with the Exploration Cohort, the performance of the models using 10 selected features are compared against the models using 10 random features. As shown in Table 1, on the validation data, the generated models resulted in higher performance metrics when 10 selected features (species) are used, as compared to the randomly generated 10 features. Especially, in terms of specificity, one can easily observe the sharp decrease to 0.35 when 10 random features are tested, as compared to the obtained specificity value of 0.85 with 10 selected species when the RF classifier is applied on the validation data. The same trend of significant decrease in terms of specificity is observed in the results of all other tested classifiers when 10 random features are tested (Table 1).

Table 1 Performance metrics of 10 identified species, compared with the 10 randomly selected species, calculated using the independent test data, Random Forest Classifier and 100-fold Monte Carlo Cross-Validation.

Independent validation dataset	
10 significant species - Mean of 100-fold MCCV	
Model	Accuracy %	Recall	Specificity	Precision	AUC	F1	
Adaboost	0.82 ± 0.08	0.85 ± 0.12	0.79 ± 0.22	0.84 ± 0.14	0.86 ± 0.07	0.83 ± 0.06	
DT	0.77 ± 0.12	0.85 ± 0.12	0.69 ± 0.28	0.77 ± 0.14	0.81 ± 0.1	0.79 ± 0.08	
LogitBoost	0.84 ± 0.07	0.85 ± 0.11	0.83 ± 0.18	0.86 ± 0.13	0.88 ± 0.07	0.84 ± 0.06	
RF	0.86 ± 0.07	0.86 ± 0.12	0.85 ± 0.16	0.88 ± 0.12	0.9 ± 0.06	0.86 ± 0.07	
SVM_opt	0.78 ± 0.1	0.85 ± 0.12	0.72 ± 0.26	0.8 ± 0.15	0.81 ± 0.09	0.8 ± 0.07	
Stack_SVM_Kmeans	0.7 ± 0.1	0.88 ± 0.12	0.52 ± 0.26	0.68 ± 0.12	0.75 ± 0.1	0.75 ± 0.06	
Stack_Logitboost_Kmeans	0.78 ± 0.1	0.85 ± 0.12	0.72 ± 0.26	0.8 ± 0.15	0.81 ± 0.09	0.8 ± 0.07	
10 random species - Mean of 100-fold MCCV	
Model	Accuracy %	Recall	Specificity	Precision	AUC	F1	
Adaboost	0.67 ± 0.11	0.92 ± 0.11	0.42 ± 0.31	0.65 ± 0.13	0.68 ± 0.12	0.74 ± 0.06	
DT	0.57 ± 0.1	0.94 ± 0.1	0.19 ± 0.28	0.56 ± 0.1	0.6 ± 0.11	0.69 ± 0.04	
LogitBoost	0.66 ± 0.11	0.93 ± 0.1	0.39 ± 0.29	0.63 ± 0.12	0.68 ± 0.11	0.74 ± 0.05	
RF	0.64 ± 0.1	0.92 ± 0.12	0.35 ± 0.3	0.62 ± 0.12	0.64 ± 0.12	0.72 ± 0.05	
SVM_opt	0.58 ± 0.08	0.96 ± 0.08	0.21 ± 0.23	0.56 ± 0.08	0.57 ± 0.12	0.7 ± 0.03	
Stack_SVM_Kmeans	0.56 ± 0.06	0.98 ± 0.05	0.14 ± 0.15	0.53 ± 0.04	0.5 ± 0.12	0.69 ± 0.03	
Stack_Logitboost_Kmeans	0.58 ± 0.08	0.96 ± 0.08	0.21 ± 0.23	0.56 ± 0.08	0.57 ± 0.12	0.7 ± 0.03	
Notes.

Bold values indicate the highest values obtained for each column (for each performance metric), and for each subtable (10 significant species, 10 random species).

As shown in Table 2, while 0.85 ± 0.05 accuracy is obtained on the Exploration Cohort, 0.86 ± 0.07 accuracy is obtained on the Validation Cohort using 10 species, RF classifier and 100-fold MCCV. With the addition of standard deviation values, the results seem quite similar and no significant difference is observed. Moreover, 0.87 ± 0.06 accuracy is obtained using 14 species on the Exploration Cohort, where four of these features (Subdoligranulum_unclassified, Ruminococcus_obeum, Lachnospiraceae_bacterium_1_1_57FAA, Lachnospiraceae_bacterium_2_1_58FAA) are not found in the Validation Cohort. Comparing 0.85 ± 0.05 accuracy value obtained using 10 species with 0.87 ± 0.06 accuracy value obtained using 14 species on the Exploration Cohort, we noticed that 4 additional features are not so necessary for improving the results.

Table 2 Performance evaluations of the 10 identified species, compared with the 14 selected species, calculated using Random Forest Classifier and 100-fold Monte carlo cross-validation, presented both for the exploration cohort and the validation cohort.

Random Forest Classifier - 100-fold MCCV	
Dataset	# of Feat.	Accuracy %	Recall	Specificity	Precision	AUC	F1	
Validation	10	0.86 ± 0.07	0.86 ± 0.12	0.85 ± 0.16	0.88 ± 0.12	0.9 ± 0.06	0.86 ± 0.07	
Exploration	10	0.85 ± 0.05	0.84 ± 0.11	0.86 ± 0.1	0.81 ± 0.1	0.89 ± 0.05	0.81 ± 0.06	
Exploration	14	0.87 ± 0.06	0.87 ± 0.1	0.88 ± 0.1	0.83 ± 0.11	0.92 ± 0.04	0.84 ± 0.06	

As compared to using all 1,331 features (0.87 ± 0.05 accuracy), those selected 10 features yielded in very similar performance in terms of accuracy and other metrics. On the Validation Cohort, it is also observed that satisfactory performance metrics are obtained using these selected species. That is to say that 10 features are sufficient to accurately classify IBD, compared with using all 1,331 features. For the diagnosis, evaluating the amounts of fewer species is more effective in terms of cost and time. Hence, those 10 features (species) shown in Table S3 are proposed as potential taxonomic biomarkers for IBD. These potential taxonomic biomarkers can be adapted to clinics for facilitating the diagnosis of IBD. If adapted to the clinic, using those 10 features, IBD diagnosis could be performed with 0.85 ± 0.05 accuracy. We thus conclude that the analysis of these 10 potential taxonomic biomarkers in gut microbiota might be used as an auxiliary diagnostic tool for suspected IBD patients.

Feature importance and their correlations

Regarding metagenome-based disease prediction, the most informative features correspond to the microbes that contribute highest to the disease prediction, which strengthens the interpretability of the model (Pasolli et al., 2016; Reiman, Metwally & Dai, 2018; Rahman & Rangwala, 2018; Duvallet et al., 2017). The identification of critical species, the ones having a key role in IBD disease development, can constitute new targets for the development of probiotics to correct microbiota aberrations (Armour et al., 2019; Gueimonde & Collado, 2012). To this end, the feature importance scores of the 14 selected species (as shown in Fig. S1) are investigated. In Fig. S1, while the Y-axis corresponds to the detected species, X-axis corresponds to their relative importance.Some of the identified species were previously reported in the literature as microbiome-associated factors for IBD. Among the top three species that are found as potential IBD biomarkers, Bacteroides xylanisolvens (Ulsemer et al., 2012) is considered as candidate next-generation probiotics, promoting gut health. Bacteroides xylanisolvens is one of the short-chain fatty acid (SCFA)producing bacteria. SCFAs are colonotrophic nutrients and they are immunoregulatory molecules (Puertollano, Kolida & Yaqoob, 2014) that may reduce pro-inflammatory cues within the gut environment. In this study, Bifidobacterium bifidum is identified as the fourth significant species. For the treatment of IBD, a recent review by Jakubczyk, Leszczyńska & Górska (2020) showed the beneficial effect of probiotics including Bifidobacterium bifidum PRL2010 and Bifidobacterium bifidum 231 strains, which are potentially involved in the etiology of IBD. Another research group demonstrated that the supplementation with Bifidobacterium bifidum causes a significant increase in IL-10 level and decrease in IL-1 levels in the colon sections, which confirmed the anti-inflammatory effect (Kumar et al., 2017). This observation verifies the anti-inflammatory effect and hence, supports the modulatory role of Bifidobacterium bifidum in terms of decreasing the inflammation, as well as the clinical symptoms of colitis. With regard to UC, the strains of Bifidobacterium bifidum are considered as promising in sustaining remission (Kato et al., 2004; Kruis et al., 2004). The abundance of Lachnospiraceae bacterium, which is identified in the top 5 important species list, has also been observed previously in IBD cohorts (Nagao-Kitamoto & Kamada, 2017).

To analyze the pairwise correlations of these selected species, SPARCC (Friedman & Alm, 2012), which is able to estimate correlation values from compositional data, is utilized. As noted by Freilich et al. (2018), SPARCC is one of the rigid algorithms that can be used to evaluate correlation in microbiome datasets. SPARCC assumes a sparse data matrix, and the ϕ (Lovell et al., 2015) and ϱ (Erb & Notredame, 2016) metrics (the published versions of which required a non-sparse matrix). These relations were illustrated in Fig. S2 using a heatmap. Between any pair of the identified species, no significant correlation is observed.

Grouping control and IBD samples via K-means clustering algorithm

IBD patients are characterized by genetically and clinically defined subgroups, which have very specific microbial compositions and functions. To this end, analysis of patient microbial metabolomic profiles could be utilized as a predictive clinical tool, which provides a foundation for personalized microbiome-based therapies (Banfi et al., 2021). This research effort also investigated whether some subgroups of IBD patients have a direct relationship with some species. In this respect, a K-means clustering algorithm is used in this study to subgroup samples. As shown in Fig. S3, three subgroups among IBD samples and four subgroups among controls were discovered. t-SNE method was employed for visualizing the identified clusters (subgroups of IBD patients and healthy samples). To this end, based on the relative abundance values of the 14 identified species, two-dimensional t-SNE maps were generated separately for (i) IBD patients subgroups, and (ii) healthy sample subgroups (Fig. 6). Cluster-colored tSNE plots were visually inspected. As shown in Fig. 6, the subgroups of IBD patients and the subgroups of healthy samples were distinct.

Figure 6 Two-dimensional t-SNE maps for (A) healthy sample subgroups, and (B) IBD patient subgroups, which are identified using K-means clustering.

Figure 7 illustrates the relative abundance values of the identified species in each of these subgroups. In Fig. 8, The presence of the six selected species was displayed more in detail for each IBD subgroup and controls. It can be concluded from both Figs. 7 and 8 that even though the samples were divided into subgroups, a single species has no direct effect on the development of IBD. Nevertheless, there are a few observations that one can make: (i) Porphyromonas asaccharolytica (shown in Fig. 8B and with grey in Fig. 7) and Peptostreptococcus anaerobius (shown in Fig. 8F and with dark blue in Fig. 7) is observed in all IBD subgroups, and not in the control subgroups; (ii) Bifidobacterium bifidum (shown in Fig. 8A and in pink in Fig. 7) is mainly observed in all IBD subgroups, it is present in very small amounts in the control subgroups; (iii) although Fig. 7 indicates that Eubacterium hallii (shown in orange) is found in all subgroups, the zoomed-in view in Fig. 8C clarifies that the relative abundance values of Eubacterium hallii is higher in IBD subgroups, compared to control subgroups (except for control 3 subgroup including 3 % of the control samples); (iv) a similar observation can be made for Lachnospiraceae bacterium 1_1_57FAA (shown in Fig. 8E and in light pink in Fig. 7) and for Dorea formicigenerans (shown in Fig. 8D and with brown in Fig. 7). It can be concluded from Fig. 7 and Fig. 8 that the relative abundance values of Bifidobacterium bifidum and Porphyromonas asaccharolytica have a role in distinguishing subgroups of IBD patients. Compared to other two subgroups of IBD patients, in the IBD_01 subgroup (including 70% of patients), the average relative abundance values are lower for Bifidobacterium bifidum, and higher for Porphyromonas asaccharolytica.

Figure 7 Relative abundance values of the identified species in healthy and IBD subgroups.

Figure 8 Zoomed-in view of the relative abundance values for: (A) Bifidobacterium bifidum, (B) Porphyromonas asaccharolytica, (C) Eubacterium hallii, (D) Dorea formicigenerans, (E) Lachnospiraceae bacterium 1_1_57FAA, (F) Peptostreptococcus anaerobius in healthy subgroups and the IBD subgroups.

As a result of the IBD patient subgroup identification steps, the enrichment of Bifidobacterium (Wang et al., 2014) and Peptostreptococcus anaerobius species in IBD patients are in accordance with previous literature(presented in detail in the Discussion section), supporting the stratification analysis performed in this study. In the future, this information could be used to stratify IBD patients more precisely and to offer more effective treatments.

Discrimination of data via principal component analysis (PCA)

To invesigate whether the samples can be divided into two clusters representing control and IBD samples, the first two and three principal components of metagenomic data were obtained using PCA as an unsupervised learning approach. Figure 9 shows that a better separation is observed between control and IBD classes when PCA is performed with the 14 selected features (Figs. 9B, 9D), as compared to using 1,331 species (Figs. 9A, 9C). Figure 9 also points out that compared to the separation using the top two principal components (Figs. 9C and 9D), the third principal component contributes to the separation of controls vs IBD samples (Figs. 9A and 9B). An interactive 3D plot for Figs. 9A, 9B, 9C, and 9D is provided as supplementary material.

Figure 9 Principal component analysis of (A, C) all IBD-associated metagenomics data, (B, D) reduced dataset that includes features for the 14 selected species, shown in 3D in (A, B) and in 2D in (C, D).

Interactive 3D plots are provided as a supplementary material.

However, it is observed that the new feature space, reduced via PCA, does not have a significant contribution in terms of classification performance. It is also important to note that since PCA maps the data into a new feature space, the original feature information is lost during this process. Thus, PCA analysis does not allow for biomarker discovery, as the species information is no longer represented in the new mapped feature space.

Hierarchical clustering of reduced IBD dataset

To better visualize the relationship between the top 14 selected species and the samples, a hierarchical clustering analysis is also conducted. The heatmap in Fig. 10 is obtained using all samples and top-scoring 14 species. Colors represent raw z-scores for samples. While the black color indicates relative abundance values just around the mean, the lighter colors denote the relative abundance values of 1 to 4 standard deviations above the mean. The first column specifies class labels, i.e., IBD patients and healthy samples are shown in red and blue, respectively. The areas that are restricted with red boxes suggest differential relative abundance values for the corresponding species in the associated subgroup. For example, while the red boxes in the 7th and 12th columns (sp9, sp6) indicate excessive levels of Lachnospiraceae bacterium_1_1_57FAA and Ruminococcus obeum in the corresponding IBD subgroups; the red boxes in the 1st, 2nd and 3rd columns (sp2, sp12, sp13) indicate excessive levels of Bacteroides xylanisolvens, Ruminococcus bromii and Subdoligranulum unclassified in the corresponding healthy subgroups. The abundance of Lachnospiraceae bacterium has been observed previously in IBD cohorts (Nagao-Kitamoto & Kamada, 2017). Kang et al. (2010) demonstrated that compared with the healthy subjects, Ruminococcus bromii was observed five times less in the fecal samples of CD patients. For intestinal microorganisms, polysaccharides and dietary cellulose are significant energy sources. R. bromii can degrade some other complex polysaccharides (involving xylan and starch) and the degradation products are known to affect the gut microbial community, as well as human health (Wang & Liu, 2020).

Figure 10 Hierarchical clustering of the samples, based on the relative amounts of the 14 selected species.

The side bar on the left hand side indicates class labels: IBD patients and healthy samples are shown in red and blue, respectively. In the heatmap, the colors represent raw z-scores. While the black color indicates relative abundance values just around the mean, the lighter colors denote the relative abundance values of 1 to 4 standard deviations above the mean. The areas that are restricted with red boxes suggest differential relative abundance values for the corresponding species in the associated subgroup.

Discussion

Recent studies have reported that commensal microorganisms play major roles in human physiology and diseases (Wang & Liu, 2020). Numerous disease states have been associated with a disturbance of the steady relationship between the gut epithelial cells and gut microbiota (known as dysbiosis) (Petersen & Round, 2014; Marcos-Zambrano et al., 2021). Among the growing literature describing disease-associated microbiota, accumulated evidence has shown that the loss of microbiota diversity is a common feature of most dysbioses (Mosca, Leclerc & Hugot, 2016). IBD is one of those diseases where dysbiosis greatly affects pathogenesis (Ungaro et al., 2019; Lazar et al., 2018). Although IBD results from a complex interplay among environmental factors, genetics, and intestinal microbiota composition (Scotti et al., 2017); major characteristic of IBD is the corruption of the interactions between the resident microbial population and host immune responses (Nagao-Kitamoto et al., 2016). In IBD patients, major shifts in gut microbial composition, such as elevated levels of facultative anaerobic pathogens and reductions in obligate anaerobic producers of short-chain fatty acids (SCFA) are mainly reported (Yoo et al., 2020). In IBD patients, while the moderations of the microbial abundances in the dysbiotic gut are observed to play critical key roles in the persistent inflammation, no single specific pathogenic species have been etiologically associated with IBD (Yoo & Kim, 2016; Scher et al., 2015). In this respect, predicting host phenotypes based on taxonomy-informed feature selection and establishing an association between microbiome and disease states could help to identify potential taxonomic biomarkers of diseases (Marcos-Zambrano et al., 2021; Maier et al., 2020).

This research effort attempts to build a classification model to aid IBD diagnosis and to discover IBD-associated bio-markers using the metagenomics data obtained from MetaHit project. Using two metagenomics datasets, which contain species as features of the human gut microbiota of IBD patients and controls, the performances of different classifiers including SVM, DT, RF, Adaboost, Logitboost were evaluated. To deal with the high dimensionality of features, some feature selection methods including FCBF, CMIM, mRMR, IG, SKB, and XGBoost were applied. The importance scores of the features were analyzed and the top-scoring species were found as related with microbiome-associated factors of IBD in literature. Among the species that were identified as potential IBD biomarkers (listed in Fig. 3), Bacteroides xylanisolvens (Ulsemer et al., 2012) and Eubacterium hallii (El Hage, Hernandez-Sanabria & Van de Wiele, 2017) were considered as candidate next-generation probiotics, promoting gut health. Other selected features of bacterial taxa, Lachnospiraceae, Parabacteroides, Blautia, Butyrivibrio, Dorea, Ruminococcus, and Roseburia were previously identified as potential biomarkers of IBD (Dubinsky & Braun, 2015). However, discovering these markers and potential therapeutic agents either require laborious wet-lab processes or they can be only discovered at the genus level. This study enabled a shotgun discovery of multiple biomarkers via applying different feature selection methods. The majority of taxa discovered in this work are associated with short-chain fatty acid (SCFA) biosynthesis in the gastrointestinal system (Wang et al., 2014; Walters et al., 2014; Nagao-Kitamoto & Kamada, 2017), which are also linked to inflammation. We believe that this finding is informative for further experimental designs in IBD research. Compared to the state-of-the-art, the proposed feature selection procedures resulted in high-performance metrics using a feature subset of smaller cardinality. Using a similar IBD-associated metagenomics dataset which is obtained from MetaHIT Project, Pasolli et al. (2016) achieved 0.80 accuracy, 0.89 AUC, 0.78 F1 score, 0.78 precision and 0.81 recall values when they reduced the number of species into 441 using Gini index. In this work, 0.88 accuracy, 0.93 AUC, 0.85 F1 score, 0.88 specificity, 0.83 precision, and 0.89 recall values were obtained by only using the 14 selected features and random forest classifier. Compared to the 0.87 accuracy, 0.92 AUC, 0.85 F1, 0.83 specificity 0.78 precision, and 0.93 recall values using all 1331 features, these 14 selected features yielded more reliable results with much lower features. These selected species could be suggested as potential IBD-biomarkers of human gut microbiota. The identified features were presented in Fig. 3 and Table S2. The associations of most of these features with IBD were also reported in the literature as follows:

There is a huge and growing literature regarding microbial gut dysbiosis in IBD, and the interplay between the immune system and microbiota in IBD. A recent review paper by Aldars-García et al. (2021) summarized the main features that are consistently found in IBD gut microbiome and their associations with the immune system. Until so far, numerous common patterns of dysbiosis in IBD, such as a decrease in bio-diversity (- and -diversity) and a deprivation of “protective” bacteria belonging to the Firmicutes phylum [12], or concomitantly increased pathogenic Gammaproteobacteria (Rajca et al., 2014) have been identified (Aden & Reindl, 2019). At the phylum level, previous work (Sheehan, Moran & Shanahan, 2015; Nishida et al., 2018; Franzosa et al., 2019) have presented imbalances in IBD patients for the Firmicutes and Bacteroidetes. In IBD patients, the decrease of the bacteria with anti-inflammatory properties, such as Bifidobacteria (Barbuti et al., 2020; Trop & Orel, 2014) has also been noted as one of the most prominent changes, in addition to the reduction of bacteria of the phylum Firmicutes (Hold et al., 2014).

On the order level of taxonomy, Enterobacteriaceae, Bacteroidales, and Clostridiales have been commonly reported across the literature as IBD bio-markers (Wingfield et al., 2018; Papa et al., 2012; Gevers et al., 2014; Morgan et al., 2012). 10 of the 14 selected microbial markers belong to the order of Clostridiales, 2 of the 14 selected features belong to the order of Bacteroidales.

At the family level, Sheehan, Moran & Shanahan (2015); Nishida et al. (2018), and Franzosa et al. (2019) demonstrated imbalances in IBD patients for the Ruminococcaceae, Veillonellaceae, Christensenellaceae, Bacteroidaceae, and Rikenellaceae. However, a recent study reported inconsistencies across studies and contradictory findings (Wu et al., 2020). Vatn et al. (2020) attempted to study the relationship between the gut microbiota composition and Crohns disease (CD), ulcerative colitis (UC). To detect taxonomic bio-markers, Vatn et al. (2020) examined fecal samples and noted that lower fluorescent signals were observed for Eubacterium hallii and Firmicutes in IBD patients, compared to the symptomatic controls (p < .05) (Vatn et al., 2020). While lower signals for Firmicutes, Lachnospiraceae, Eubacterium hallii, and Ruminococcus albus/bromii were observed for UC patients compared to the symptomatic controls; higher signals were detected for Bacteroides fragilis (Vatn et al., 2020). Duranti et al. (2016) constructed a murine model of colitis via inducing TNBS (2,4,6-trinitrobenzene sulfonic acid). Using this model, they reported that oral supplementation with Bifidobacterium bifidum slowed down colonic edema, macroscopic damage, histological scores, and prevented weight loss Durantietal2016. This finding is also in line with the observation that depleted biosynthesis of short-chain fatty acids (SCFA) might be associated with IBD pathogenesis. Surana & Kasper (2017) reported that Lachnospiraceae species offer protection against colitis . In another study, Morell Miranda, Bertolini & Kadarmideen (2019) showed that Crohns disease patients with no detectable sign of depression exhibited higher abundances for Bacteroides xylanisolvens, compared with Crohns disease patients with signs of depression (Morell Miranda, Bertolini & Kadarmideen, 2019). Swidsinski et al indicated that the levels of Ruminococcus bromii have changed in IBD patients (Swidsinski et al., 2005). Using a customized phylogenetic microarray (Kang et al., 2010) demonstrated that in healthy persons , some types of bacteria which belong to the Firmicutes phylum including Eubacterium rectale of the Lachnospiracea and Ruminococcus albus, R. callidus, R. bromii, and F. prausnitzii of the Ruminococcaceae were identified 5 to 10 times more, compared with the Crohn’s disease patients (Forbes, Van Domselaar & Bernstein, 2016). Franzosa et al. (2019) noted Dorea formicigenerans and Ruminococcus obeum which show the strongest enrichments in non-IBD controls. In the same study, the authors presented metagenomically contributed enzymes that were differentially abundant in IBD, annotated by their taxonomic contributors. Bifidobacterium bifidum, Ruminococcus bromii, Ruminococcus obeum, Coprococcus sp, Lachnospiraceae bacterium were listed as taxonomic contributors. In another study, using a linear model and controlling for subject age and medication use, (Franzosa et al., 2019) revealed 50 microbial species, which were differentially abundant in IBD relative to controls. Coprococcus comes, which was one of the 14 selected species, was also included in their list. Similarly with these findings, 14 selected features of this study include Lachnospiraceae bacterium_1_1_57FAA, Lachnospiraceae bacterium_2_1_58FAA, Eubacterium hallii, Ruminococcus bromii, Bacteroides xylanisolvens, Bifidobacterium bifidum, Dorea formicigenerans and Ruminococcus obeum; and these species are suggested as potential taxonomic biomarkers of IBD. Among these species, Coprococcus Comes and Eubacterium Hallii are both known to be propionate-producing bacteria, whereas Dorea formicigenerans is a phylogenetically related species. Propionate is a health-promoting short-chain fatty acid (SCFA) with inflammation suppression properties, and its depletion has been linked to inflammatory bowel disorders recently (Reichardt et al., 2014; Engels et al., 2016; Taras et al., 2002; Louis & Flint, 2017). Therefore, the findings presented in this study might be an implication of an anti-inflamatory cluster discovery associated with IBD.

Lloyd-Price et al. (2019) found that the bile acids, which were altered chemically modified by the gut microbes, were spoiled during IBD, in tandem with molecular regulation in groups of microbes. These gut microbes involve a set of bacteria belonging to the Subdoligranulum genus that is found in nearly everyone but are consumed during inflammation. These bacteria have not been isolated or characterized previously (Lloyd-Price et al., 2019). Subdoligranulum was one of the 14 selected species.

In a mouse model of colorectal cancer, Tsoi et al. found that compared with non-tumor tissues, in human colon tumor tissues and adenomas, the levels of Peptostreptococcus anaerobius are elevated. They observed that these bacteria expand colon dysplasia. On colon cells, P. anaerobius acts on TLR2 and TLR4 and raises the levels of reactive oxidative species, which accelerates cholesterol synthesis and cell proliferation (Tsoi et al., 2017). Similarly, in the pathogenesis of IBD, increasing evidence has recently highlighted immune-system dysfunction, especially toll-like receptors (TLRs)-mediated innate immune dysfunction, as central players in the pathogenesis of IBD. The expression of TLR2, 4, 8, and 9 genes are upregulated in patients with active UC (Lu et al., 2018). Peptostreptococcus anaerobius was one of the 14 selected species. As shown in Fig. 8F and with dark blue in Fig. 7, the abundance values of Peptostreptococcus anaerobius is higher in all IBD patient subgroups compared to control subgroups.

To maintain our intestinal homeostasis and its functions, a fragile balance between the intestinal immune system and the gut microbiota needs to preserved (Maslowski & Mackay, 2011). The studies using animal models and cutting-edge research conducted on clinical patients have both emphasized the crucial role of the intestinal microbiota in terms of developing IBD, its progression, and specifying the severity of IBD (Becker, Neurath & Wirtz, 2015; Aldars-García et al., 2021). An increasing number of microbiome studies paves the way to the advancement of microbiome-based biomarkers as useful disease indicators (Aldars-García, Chaparro & Gisberat, 2021). Additionally, the reconstruction of the composition and the diversity of the commensal microbiota became popular as a novel therapeutic intervention to adjust the microbial imbalance involved in IBD development and progression (Zhang et al., 2017). Along this line, the findings of this study support the idea that certain species might activate intestinal inflammation and hence may lead to IBD pathogenesis. These findings not only pave the way to the identification of novel diagnostic taxonomic bio-markers but also accelerate anti-bacterial drug development studies for the treatment of IBD (Ungaro et al., 2019). In summary, this research effort provides the foundation for a framework to be developed in the future for precise biomarker discovery, which enables targeting a minimal number of diagnostic/therapeutic markers with large effect sizes.

Conclusions

Gut microbiota can act upon the host immune system and metabolism, which are central to organizing several aspects of host activities. The perturbations of the gut microbiota are known as the gut dysbiosis, and they are observed in several diseases including colorectal cancer, food allergies, infection and inflammations in the gut such as inflammatory bowel disease (IBD) (Zeng, Inohara & Nuñez, 2017). Although IBD has a complex etiology, involving immune dysregulation, genetics and environmental factors, it has been consistently shown that there is a disease-dependent restriction of biodiversity and an imbalanced gut microbiota composition in IBD patients (Aldars-García, Chaparro & Gisberat, 2021). Along this line, the metagenomic analysis of human microbiome allows us to reveal several important phenotypical signals, mainly disease states, since the microbiome is regulated via human-microbiome symbiosis (Malla et al., 2019). In IBD, the precision of the diagnosis is important to prompt an effective treatment, and hence there is an utmost need to develop a novel classification technique that can expedite IBD diagnosis. This study utilizes several supervised and unsupervised machine learning algorithms on IBD-associated metagenomics data to aid diagnostic accuracy, to investigate potential pathobionts of IBD, and to find out which subset of microbiota is more informative than other taxa applying some of the state-of-the-art feature selection methods. Overall, this paper provides justification for the use of advanced feature selection and machine learning techniques on disease-associated metagenomics data. With this study, we hope to illuminate the gut microbiome functions that constitute a healthy microbiome; and we hope to accelerate the identification of potential targets for microbiome-based diagnostics and therapeutics.

Supplemental Information

Supplemental Information 1 Performance evaluations of different classifiers on IBD metagenomics dataset, utilizing 100-fold Monte Carlo Cross Validation and using (i) all features (without feature selection); (ii) top 100 features selected using CMIM, mRMR, FCBF, SKB, IG and XGBoos

Click here for additional data file.

Supplemental Information 2 The scaled importance values of top 100 features, calculated using CMIM, mRMR, FCBF, SKB, IG and XGBoost feature selection methods across different classifiers

Click here for additional data file.

Supplemental Information 3 10 species that are found in the Validation Cohort out of the 14 species identified on the Exploration Cohort

Click here for additional data file.

Supplemental Information 4 Feature importance scores of the 14 selected species for IBD-associated metagenomics dataset (averaged over Select K Best (SKB), Information Gain (IG) and XGBoost feature selection methods, across different classifiers). The scaled importance values of to

Click here for additional data file.

Supplemental Information 5 Pairwise correlation heat map of 14 selected features. Correlation values are calculated using SPARCC method, which is capable of estimating correlation values from compositional data and hence it is one of the rigorous approaches that can be applied to a

Click here for additional data file.

Supplemental Information 6 Selection of the optimum number of the clusters for (A) controls and (B) IBD patients. Using the elbow method, four and three clusters were found as the optimum number of clusters for the controls and the IBD patients, respectively

Click here for additional data file.

Supplemental Information 7 An interactive view of the top two principal components of the reduced dataset that includes features for the 14 selected species

Click here for additional data file.

Supplemental Information 8 An interactive view of the top two principal components of all IBD-associated metagenomics data

Click here for additional data file.

Supplemental Information 9 An interactive view of the top three principal components of the reduced dataset that includes features for the 14 selected species

Click here for additional data file.

Supplemental Information 10 An interactive view of the top three principal components of all IBD-associated metagenomics data

Click here for additional data file.

Additional Information and Declarations

Competing Interests

Author Contributions

Data Availability

Burcu Bakir-Gungor is an Academic Editor for PeerJ.

Burcu Bakir-Gungor conceived and designed the experiments, analyzed the data, authored or reviewed drafts of the paper, and approved the final draft.

Hilal Hacılar conceived and designed the experiments, performed the experiments, analyzed the data, prepared figures and/or tables, authored or reviewed drafts of the paper, and approved the final draft.

Amhar Jabeer performed the experiments, prepared figures and/or tables, and approved the final draft.

Ozkan Ufuk Nalbantoglu conceived and designed the experiments, performed the experiments, analyzed the data, authored or reviewed drafts of the paper, and approved the final draft.

Oya Aran conceived and designed the experiments, analyzed the data, prepared figures and/or tables, authored or reviewed drafts of the paper, and approved the final draft.

Malik Yousef conceived and designed the experiments, prepared figures and/or tables, and approved the final draft.

The following information was supplied regarding data availability:

The IBD-associated metagenomics dataset is available at MetaHIT Project and EBI (ERA000116)

- https://www.gutmicrobiotaforhealth.com/metahit/

- https://www.ebi.ac.uk/ena/browser/view/ERA000116

The codes are available at GitHub, Weka, and scikit-learn:

- https://github.com/dmlc/xgboost

- https://weka.sourceforge.io/doc.packages/fastCorrBasedFS/weka/attributeSelection/FCBFSearch.html

- https://www.cs.waikato.ac.nz/ ml/weka/index.html (CMIM, mRMR)

- https://scikit-learn.org

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
