# Peer review of "Inflammatory bowel disease biomarkers of human gut microbiota selected via different feature selection methods"

_PeerJ, doi:10.7717/peerj.13205_

## Round 0.1 · original submission · Major Revisions

The reviews provided are detailed and constructive. All three reviewers have expressed major concerns, especially with possible model overfitting. Reviewer 1 and Reviewer 2's suggestion that the findings should be validated on a larger open-sourced dataset needs to be implemented.

·

Basic reporting

• The writing is clear with a few small mistakes in places
• The literature review is missing some important papers in the area (details provided in following sections)
• Strong claims about elements of the microbiome causing disease are made (lines 17 – 18, 39 -42). For the majority of cases only associations have been found, direct causation is rarely confirmed. Strong claims about causation should have strong evidence provided.
• The aetiology of IBD is complex and many other factors are associated with IBD, not just the gut microbiome (lines 19 – 20). This should be made clearer.
• Figure 5 should have y axis labels and confidence intervals to allow proper comparison of models
• Figure 6 C is impossible to read and understand, it’s too small
• Figure 7 is missing x and y axis labels
• Figure 8 is missing y axis labels, so I don’t know what it’s measuring (relative abundance?). There are too many groups and it’s difficult to tell species apart. I recommend visualising only the 10 or 12 species most abundant and combining the others into an “Other” category
• Figure 9 A) B) is very difficult to understand from a PDF (it’s 3D). I recommend including an interactive 3D plot in the supplemental instead.

Experimental design

• The methods section doesn’t describe the process of processing raw MetaHit sequence data to create the metagenomics dataset. Were the sequence data quality checked with FastQC and multiQC? Were non-biological sequences (e.g. adapters) removed with trimmomatic? Metaphlan2 uses a read mapping approach to match against reference genomes, so removing non-biological sequences is important. It can’t be assumed that these steps were done by the MetaHit team that uploaded their data.
• The methods section doesn’t describe preprocessing done to the metagenomic data. Different samples have a different number of bacterial reads, how was this normalised? See doi.org/10.1371/journal.pcbi.1003531 and doi.org/10.1186/s40168-017-0237-y
• What criteria is used to define if a patient has inflammatory bowel disease? (E.g. the Montreal classification) Were the patients treated with antibiotics? This will change the structure of the microbiome significantly. Was IBD in an active or remissive state when the samples were taken? I think a more thorough description of patient demographics is required – perhaps a table?
• Unfortunately, from Figure 2 it appears the feature selection process is fatally flawed, causing all estimates of predictive power to be overstated. Feature selection is done on the entire metagenomics dataset, and models are trained on the reduced dataset using cross-validation. Feature selection should instead be included inside the resampling method to estimate the stability of the feature subset and avoid overfitting. Ambroise and McLachlan (2002; doi.org/10.1073/pnas.102102699) and Svetnik et al (2004; doi.org/10.1007/978-3-540-25966-4_33) showed that improper use of resampling to measure performance will result in models that perform poorly on new samples.
• Spearman rank correlation should not be used on metagenomic data because it is compositional (the sum of all bacterial proportions in each sample is constrained to 1), see 10.1371/journal.pcbi.1002687, doi.org/10.3389/fmicb.2017.02224
• Comparing model performance should involve analysing confidence intervals generated from cross-validation resamples and using a statistical approach
• I’m not sure what the unsupervised approach contributes significantly to the paper. From Figure 10 there are two small clusters but there are many confounders that could contribute to this (e.g. antibiotic use or other treatments). To understand the relationship between samples and species other methods could be used such as interpretable machine learning (e.g. SHAP arXiv:1705.07874)

Validity of the findings

• The flawed feature selection procedure used means all prediction estimates are invalid. They should be recalculated with the correct feature selection procedure, incorporating feature selection inside of the cross-validation resampling process
• Given the performance claims made about the models (AUROC approaching 1 in Figure 4) I think it is very important that the trained models are externally validated. There are many large metagenomic datasets available such as Gevers et al. 2014 which contains more than 1000 samples (10.1016/j.chom.2014.02.005). Previous machine learning work predicting IBD has used external validation to demonstrate the ability of trained models to generalise (10.1109/TCBB.2018.2831212). Qiita (https://qiita.ucsd.edu) is an excellent platform to browse and discover new microbiome datasets

When using external validation data it's important that the data can be harmonised across data sources (e.g. species or genus names match, and the same reference database is used)

·

Basic reporting

This article is generally well written, uses a professional academic structure, and is relatively self-contained. The argument presented is clear and uses professional English.

Further supporting information is required for the claims made in lines 38-40.

There are some minor spelling and grammatical errors including:
Line 60: referencing style should be kept consistent throughout.
Line 297: Euclidean (E not e).
Line 308: Remove ?
Lines 325-326: Grammatical error "applying a careful feature selection". Please rephrase.

There are some difficulties with the following Figures:
Figure 4: Please use . instead of , for the decimal point. Please also add (%) as units to the header of the Accuracy columns. Increasing font size would be desirable here even if it means increasing the table sizes and significantly reducing the white space between them.
Figure 5: Colours are difficult to differentiate for those that are color blind. Please consider differentiating the bars with different fill textures/patterns. Please also add a label for the y-axis on each subfigure and please include confidence intervals.
Figure 6: The text in all three sub-figures is illegible. Note the x-axis label for (a) has been cut off by sub-figure (c). The numeric values in (c) are completely illegible and should be removed to aid clarity.
Figure 7: Please label y-axis. lease increase font size on x-axis. Please adjust the y-axis so that the numeric values are consistent on both.
Figure 8: Please label y-axis and increase the font size of the text on the right-hand side.
Figure 9: Please adjust the yellow coloring as it is difficult to see with the white background.
Figure 10: Perhaps this is best presented as a whole page image in landscape format?

Experimental design

The aims of the paper clearly fall in the remit of the journal.

The research question is well defined, relevant and meaningful. The authors have described how they feel the results address the deficiencies of the knowledge base.

There are however some concerns over the scientific methodology.
- The feature selection approach in its current form is at risk of leading to overfitting of the classification models. The feature selection process should be carried out as part of the resampling method. The methodology should be revised and the experimentation repeated once this change has been implemented.
- More information is required on the pre-processing of the raw MetaHit sequence data. Are there any further preprocessing steps conducted to refine the metagenomic data?
- How were patients classed as having (or not having IBD)? How homogenous was the IBD patient cohort (and indeed the control cohort)?

- Line 218: How were the parameters optimized? In general, please provide parameters and where appropriate the names of functions used from toolboxes.
- Line 220: Please confirm if the train/test split was consistent across all models.
- Line 222: The intersection of features would also be interesting to examine as these are the features that consistently came up - are they the most useful?

Some minor points:
Lines 126-127: please include a citation to justify the claim "commonly used as a feature".
Lines 128-139: Please link back to the three types of features discussed in lines 122-124. Lines 122-124 are introduced but appear to be unused in the later discussion.

Please clearly explain what the 3302 columns in Figure 1 represent. I would also recommend in Figure 1 moving the "no of sick/healthy samples" from the right-hand side to the left-hand side so that we can see the 328 samples are split into 148 and 234. Currently, it looks like the text is associated with the 1st and 2nd row of data explicitly.

Validity of the findings

The primary concern here follows from the risk of overfitting caused by the methodological approach (as described in my comment above). The results should be revisited once the methodology is revised. There are some general comments which would be beneficial for the authors to consider in their revision.

Whilst the sample size is reasonable for this study, any findings from this work (with the revised methodology) would be significantly strengthened by validating the findings on a larger open-sourced dataset. Whilst this is substantial work it will result in more robust findings, support model comparisons in the future, and ease reproducibility, which will hopefully see the author's efforts rewarded through increased community adoption of their approach.

Line 250-251: Please comment on the biological relevance of the features contained in the intersection of all feature selection methods. Is XGBoost good because it identifies particularly interesting species or are the really interesting species contained within the 10 species of the intersection?
Line 270: Please elaborate on how the values 12 and 7 were identified. They are not immediately obvious from Figure 7.
Lines 263-281: Please comment on the role the isolated species are known to have in biology similar to the discussion presented in lines 258-262.
Line 298: Why cut off at the top 24 species? This seems arbitrary without further explanation.

Discussion Section:
Lines 325-326: The description of "careful feature selection" is perhaps misleading. The authors apply a number of feature selection methods, but what is the take-home message for the reader? Which feature selection method should be used? Is it important to select a particular feature selection method depending on the classification model to be used? Should a number of feature selection methods be used and the final features are determined by the union/intersection?
Lines 330 and 334: "unprecedented performance/results" - This is a bold statement and should be revised. The conclusion "feature selection can improve classification results" is a well-established finding so there is precedence that your results should be higher than previous studies. The performance increase is certainly good but I don't think unprecedented is the appropriate description.
Line 339: "leads to a framework" - perhaps the rephrase along the lines of "provides the foundation for a framework to be developed in future work" would be more appropriate.
Line 341: "would imply a potential for narrowing" - I don't think it is clear what is meant here. I think a rephrase would be helpful for the reader.
Line 352: "provides a blueprint". I think one of the following may be more appropriate: "provides evidence" or "provides justification" or "provides motivation" or "provides support".

Additional comments

The paper is well written and I would compliment the authors for some good practices, such as Figures 2 and 3 which clearly and concisely illustrate the concept outlined in the main discussion. The paper presents promising results however, further experimentation should be conducted to ensure the classification models are not inadvertently overfitted as a result of the feature selection approach outlined in the current manuscript.

·

Basic reporting

The reporting of this article is, to describe it briefly, unbalanced. It is written in clear, professional English, and there are sections of well developed and explained text, both in Materials and Methods and in the Experiments (according to PeerJ standards it should be named Results), but there are other sections that are clearly missing content, or the one written is contradicting itself. As an example, we do not need to look further than the Abstract, where we can find a claim that dysbiosis in the cause of IBD and other gastrointestinal illnesses, to a few lines later read that we do not have a clear picture of how this dysbiosis affects IBD, just that they show a strong association (this is also repeated in the Introduction). Claims of causality should be avoided unless evidence of this relationship is abundant, and with the gut metagenome we are just not there yet.
Getting into the Introduction, I find that some general description of what we know about the relationships between IBD and dysbiosis gut environments would be beneficial to highlight the dire need for this kind of diagnosis approaches. I would add a brief description of the most common symptoms of IBD and the relationship of between this illness and the gut to lines 51 to 57. The mention of the high comorbidity of IBD with other dysbiosis related illnesses, as depression, anxiety or obesity could also highlight the relevance of this research.
Regarding the figures, I do not have any major complaints. The tables in Figure 4 may be too small in some screens, so maybe they should be independent supplementary tables. Other than that, they are clear, well described and easy to interpret.
Raw data was supplied through the project’s website. This should still be disclosed on the paper.

Experimental design

The experimental design follows the trend of using different Machine Learning algorithms for classification of different atypical markers of illnesses. This is a powerful approach both to understand better which alterations in the microbiotic environment are most influential on IBD, and for clinical purposes as diagnostic and treatment.
The research question is well defined and relevant, and the investigation is, as far as I can see, performed according to technical and ethical standards.
In the description of the methods we find, again, this unbalance that we previously saw in the introduction and abstract. The first part of that section (lines 122-144) and the Feature Selection descriptions are good. Model Selection and Unsupervised Learning, however, are too brief and shallow. There is no description of what any of the classifiers used, nor a justification of their use. In fact, the Model Construction section is composed mostly of a description of the performance metrics used to analyze how well those algorithms performed, instead of why those algorithms were used and how they work. Something similar happens with the Unsupervised learning, where the description is so brief that a description of the algorithms has been included in the Experiments (Results).

Validity of the findings

To my knowledge the results presented in this paper are novel and relevant. I would like the authors to take a bit of time to try to link them to other research in IBD and metagenomics, mostly on the topics I already mentioned on my comments on Basic Reporting. IBD is a complex, multifactorial and comorbid illness, and that should be addressed both in the introduction and description of the issue, and on the discussion and interpretation of our results, even if the scope of our analysis does not include all those factors or comorbid conditions.
I must admit that I do not like to see some of the results, as the most relevant species for IBD diagnosis according to your analysis, firstly mentioned on the Discussion. Maybe restructuring the article according to PeerJ Standard Sections template would help avoiding this. I understand that the focus of this paper is on the use of Machine Learning as a diagnostic tool, but if you are going to mention it at all, it still should be mentioned in the Results section.

---

## Round 0.2 · Minor Revisions

The revised version of the manuscript has been seen by one of the previous reviewers. They acknowledge the improvements, but the Reviewer still has several minor concerns that need to be addressed.

·

Basic reporting

The use of "our", "we" throughout the manuscript makes the writing quite informal and I would recommend that this be changed throughout. The manuscript is the presentation of a scientific investigation. It is not owned by, nor belongs to, any set of authors as the same study can be proposed and implemented by others. Therefore, it is more appropriate to refer to "The study", or "the results" and what explicitly occurred during the investigation, rather than "Our study" "Our dataset", "we applied" etc..

The section "Validation on external data" includes the use of , instead of . for decimal points and should be changed.

References are generally appropriate and in place. Page 9, websites should be referenced properly alongside the other references (including the date the website has been accessed on since webpages can change).

Figure 1: No of sick/healthy samples should be No. of ... or # sick/healthy samples.

Table 1: There are a lot of values. Perhaps it would be helpful to highlight in bold the highest accuracy/recall/specificity etc. in each sub-table and the lowest std. dev. .

In the Dataset and Preprocessing Steps section (and elsewhere as appropriate) all weblinks should be properly referenced in the list of citations including date on which the content was accessed since the contents of the weblinks can be changed.

Changes in Figure 4 mean that the units for Accuracy are no longer %.

Please clarify to which dataset the results in Figure 5 belong to.

Figure 7: "Relative abundance" ....relative to what? Y-axis now has values but remains unlabelled and without units.

Figure 8:Y-axis is still not labelled/no units provided.

Figure 10 is still difficult to read although is more legible from the original submission. I think the authors should reflect on the "so what?" question here: What do you want the reader to take away from this figure? There is a lot of information presented in Figure 10. If you feel the reader needs all of this information in order to understand the conclusions drawn then please make the figure larger and more legible. Perhaps highlight specific regions of the figure and label with (a), (b), (c) etc to assist with your discussions. Alternatively, the authors may wish to revise the figure to present the key information (i.e. less information) more clearly to the reader so that they can interpret the results and follow the conclusions discussed in the paper. You can generate a Hierarchical clustering map...this doesn't mean you should.

Experimental design

Line 442: "Very similar preprocessing"...How similar? What was the same? What was different and why are the differences insignificant?

In the validation on external data RF is isolated however the initial findings from Table 1 would suggest LogitBoost would be equally as strong or appropriate to use. The authors may wish to reiterate why they focus on RF explicitly.

The inclusion of the independent dataset has improved the rigour of the study.

The research is original and of interest to the audience of the journal.

Validity of the findings

Line 424 "As shown in Figure 5..." These findings can also be found in Table 1.

Table 1 has a lot of values. Perhaps it would be helpful to highlight in bold the highest accuracy/recall/specificity etc. in each sub-table and the lowest std. dev. .

In Table 1 it is worth commenting that CMIM, FCBF and MRMR show signs of poor fitting across all models with low accuracy and high recall. (re lines 411-413).

Results Section (from line 419-...) RF is singled out here but it could be argued LogitBoost performed best (or as good as RF) when all 1331 features are used.

Line 431-432: "diagnosis could be performed with 88% accuracy" - so what? Is good enough for clinical adoption? At present this sentence just repeats what is in the table but doesn't really make a statement. The subsequent "only analysing the amounts of 14 specific species" - More appropriate to say "14 features are sufficient" to accurately classify IBD compared with using all 1331 features. (Note the typo: "by checking only analyzing".

Line 433-434: I think the final sentence of this section needs to be rephrased, perhaps purpose rather than proposed. Also the subsequent Validation on external data seems to conflict with this point since it appears that only 10 species are required. If line 434 is a hypothesis then test this on the independent external dataset (which turns out to not be fully possible - hence explain why and possibly revise under which circumstances people would use the 14 species). Furthermore given that 10 species are highlighted in the validation dataset it might be appropriate to leave any conclusion/propositions until after all the results are presented.

Validation on External data section: There are references that the results are shown in Table 1 but I was unable to see these findings - perhaps a table is missing?

Line 455-459: This appears to be contradictory. 10 selected species yielded higher performance metrics in the validation than exploration. Therefore the 4 additional features are beneficial? Please clarify - are you saying, that the 10 species from the validation cohort considered in isolation in the exploration cohort are less useful than 14 features in the exploration cohort? Furthermore, using the 10 selected species in both cohorts it was found that the exploration cohort was easier to classify than the validation cohort - in which case why might this be the case?

Additional comments

As a side note, it would be appreciated if the authors could cross-reference where the changes can be observed in the revised manuscript in their responses.

---

## Author Rebuttal · Round 0.2

# Responses to Reviewers for 57696

# Inflammatory bowel disease biomarkers of human gut microbiota selected via ensemble feature selection methods

Burcu Bakir-Gungor[1], Hilal Hacilar, Amhar Jabeer, O. Ufuk Nalbantoglu, Oya Aran, and Malik Yousef

## I. Rebuttal Letter Style:

We first thank all the reviewers for their extremely thoughtful suggestions and their valuable time in reviewing our paper. We are very excited to have been given the opportunity to revise our manuscript. In this revised version, we list all detailed **comments from each reviewer in bold**, followed by our corresponding responses and revisions. As an attachment to this rebuttal letter, we have also given our revised manuscript as a marked-up copy and the final version of our revised manuscript. We also would like to note that the page numbers that we refer to in this rebuttal letter refer to the page numbers in the marked-up copy of our revised manuscript.

## Editor comments (Joao Setubal)
• **The reviews provided are detailed and constructive. All three reviewers have expressed major concerns, especially with possible model overfitting. Reviewer 1 and Reviewer 2's suggestion that the findings should be validated on a larger open-sourced dataset needs to be implemented.**

We thank all the reviewers for their detailed and constructive comments. In our revised study, we have redesigned our experiments in order to prevent model overfitting. Based on Reviewer 1 and 2's suggestions, we have validated our findings using an independent test set. In the revised manuscript, we have rewritten all sections, redrawn the figures, and added several new references including review papers well known in this area. In the discussions section, we discussed the gut bacteria that were identified as potential taxonomic biomarkers of IBD, and we presented more detailed information about the relationship between these bacteria and IBD via referring to the literature.

## II. Responses to Reviewer #1 (B Wingfield):
### Basic reporting

• **The literature review is missing some important papers in the area (details provided in following sections).**

We thank the reviewer for pointing out this issue. In the revised manuscript, we have rewritten several sections including Introduction, Materials, Methods, Discussions, and added several new references including review papers well known in this area.
* * *
[1] Correspondance: burcub@gatech.edu

**Strong claims about elements of the microbiome causing disease are made (lines 17 – 18, 39 -42). For the majority of cases only associations have been found, direct causation is rarely confirmed. Strong claims about causation should have strong evidence provided.**

We thank the reviewer for this suggestion. We have edited the sentences in the above-mentioned lines and removed our strong claims about elements of the microbiome causing disease.

**• The aetiology of IBD is complex and many other factors are associated with IBD, not just the gut microbiome (lines 19 – 20). This should be made clearer.**

We thank the reviewer for pointing out this issue. In the revised manuscript, other factors that are associated with IBD are also emphasized in the abstract, introduction and discussion sections.

**• Figure 5 should have y axis labels and confidence intervals to allow proper comparison of models.**

We thank the reviewer for pointing out this issue. In the revised manuscript, we have performed several experiments as suggested by the reviewers and generated new Figures such as Figure 4 that display confidence intervals as well. In the revised manuscript, the mean and standard deviation values are also reported in Supplementary Tables 1 and 4. The y axis on Figure 4 of the revised manuscript indicates the accuracy(%), Area Under Curve, F1-measure values, as labelled using different colors.

**• Figure 6C is impossible to read and understand, it's too small.**

We thank the reviewer for pointing out this issue. Based on the reviewer's suggestion in "Experimental Design" part, in the revised manuscript, in order to calculate feature correlations, instead of using Spearman rank correlation, we have utilized SPARCC (Friedman and Alm, 2012), which is capable of estimating correlation values from compositional data. As noted by (Gloor et al, 2017), SPARCC is one of the rigorous approaches that can be applied to analyze correlation in microbiome datasets. SPARCC assumes a sparse data matrix, and the $\phi$ (Lovell et al., 2015) and $\rho$ (Erb and Notredame, 2016) metrics (the published versions of which required a non-sparse matrix). In the revised manuscript, we have redrawn Supplementary Figure 2 in a more readable format.

**• Figure 7 is missing x and y axis labels**

We thank the reviewer for pointing out this issue. In the revised manuscript, we have performed several experiments as suggested by the reviewers, redrawn Supplementary Figure 3 and added x and y axis labels.

**• Figure 8 is missing y axis labels, so I don't know what it's measuring (relative abundance?). There are too many groups and it's difficult to tell species apart. I recommend visualising only the 10 or 12 species most abundant and combining the others into an "Other" category.**

We thank the reviewer for the suggestion. It is true that Figure 8 is measuring the relative abundance; and we add it as y axis label. As described more in detail in the revised manuscript,

we have created patient and healthy subgroups based on k-means clustering. Based on the reviewer's comment, in the Figure 7 of the revised manuscript we have visualized 14 identified species and in Figure 8 we have visualized the most informative species.

• **Figure 9 A) B) is very difficult to understand from a PDF (it's 3D). I recommend including an interactive 3D plot in the supplemental instead.**

We thank the reviewer for the suggestion. Based on the reviewer's comment, in the revised manuscript we have included an interactive 3D plot for Figure 9.

**Experimental design**

• **The methods section doesn't describe the process of processing raw MetaHit sequence data to create the metagenomics dataset. Were the sequence data quality checked with FastQC and multiQC? Were non-biological sequences (e.g. adapters) removed with trimmomatic? Metaphlan2 uses a read mapping approach to match against reference genomes, so removing non-biological sequences is important. It can't be assumed that these steps were done by the MetaHit team that uploaded their data.**

Thanks for raising this point. Prior to taxonomic analysis with MetaPhlAn, we followed the standard quality analysis, which is proposed by the Human Microbiome Project (HMP)[2]. According to that, firstly the duplicate reads were removed by a modified version of EstimateLibraryComplexity in Picard tool package. The quality trimming was performed using TrimBWAStyle script[3], which is also recommended by the HMP SOP. We skipped the human genome contamination removal step, as the data provided is already free of human contamination. Prior to taxonomic analysis, we also filtered out the reads, which are smaller than 90bp in length. We have added this explanation to the revised manuscript.

• **The methods section doesn't describe preprocessing done to the metagenomic data. Different samples have a different number of bacterial reads, how was this normalised? See doi.org/10.1371/journal.pcbi.1003531 and doi.org/10.1186/s40168-017-0237-y**

We thank the reviewer for raising this point. We have considered the standard relative abundance normalization by dividing the read count of each taxonomic bin to the total number of the reads for a sample, so that taxonomic abundances are real numbers in the range of [0, 1], which sums up to 1 within each sample. Samples containing less than 1 million reads were discarded. We have added this explanation to the revised manuscript.

• **What criteria is used to define if a patient has inflammatory bowel disease? (E.g. the Montreal classification) Were the patients treated with antibiotics? This will change the structure of the microbiome significantly. Was IBD in an active or remissive state when the samples were taken? I think a more thorough description of patient demographics is required – perhaps a table?**
* * *
[2] Human Microbiome Project Consortium. Structure, function and diversity of the healthy human microbiome. Nature 2012;486(7402):207–214. PMID: 22699609. https://www.hmpdacc.org/hmp/doc/ReadProcessing_SOP.pdf

[3] https://github.com/dgpinheiro/bioinfoutilities/blob/master/TrimBWAStyle.pl

In the original study that we obtained the metagenomics data (PRJEB2054), it is stated that: "Healthy controls were recruited among family relatives of IBD patients; antibiotic treatment for at least 4 weeks before fecal sample collection was excluded. IBD subjects were in clinical remission for at least 3 months, and had stable maintenance therapy with mesalazine or azathioprine."

The patient demographics providing the country, gender, age, body-mass index are available in the original study. In our revised manuscript, we added such a sentence.

• **Unfortunately, from Figure 2 it appears the feature selection process is fatally flawed, causing all estimates of predictive power to be overstated. Feature selection is done on the entire metagenomics dataset, and models are trained on the reduced dataset using cross-validation. Feature selection should instead be included inside the resampling method to estimate the stability of the feature subset and avoid overfitting. Ambroise and McLachlan (2002; doi.org/10.1073/pnas.102102699) and Svetnik et al (2004; doi.org/10.1007/978-3-540-25966-4_33) showed that improper use of resampling to measure performance will result in models that perform poorly on new samples.**

Thanks for the point. In the revised version, we have splitted the data as training data and test data. We applied the feature selection algorithms on the training part. We tested our classification methods on the testing part. Actually, we have performed 100-fold Monte Carlo Cross Validation (MCCV) and the average of all performance metrics are calculated.

We have updated the manuscript and the results presented in Tables 1-2, Supplementary Tables 1-3 and all Figures. We have also updated Figure 2 that illustrates the workflow of our method. In the revised Figure 2, we showed this splitting process.

• **Spearman rank correlation should not be used on metagenomic data because it is compositional (the sum of all bacterial proportions in each sample is constrained to 1), see 10.1371/journal.pcbi.1002687, doi.org/10.3389/fmicb.2017.02224**

Based on the reviewer's suggestion, in the revised manuscript, in order to calculate feature correlations, instead of using Spearman rank correlation, we have utilized SPARCC (Friedman and Alm, 2012), which is capable of estimating correlation values from compositional data. As noted by (Gloor et al, 2017), SPARCC is one of the rigorous approaches that can be applied to analyze correlation in microbiome datasets. SPARCC assumes a sparse data matrix, and the $\phi$ (Lovell et al., 2015) and $\rho$ (Erb and Notredame, 2016) metrics (the published versions of which required a non-sparse matrix).

• **Comparing model performance should involve analysing confidence intervals generated from cross-validation resamples and using a statistical approach.**

Thanks for the point. In the revised version, we have added confidence intervals, as shown in Figure 4. We have performed 100-fold Monte Carlo Cross Validation (MCCV) and the average of all performance metrics are calculated. In the revised manuscript, the mean and standard deviation values are also reported in Supplementary Tables 1 and 3.

**• I'm not sure what the unsupervised approach contributes significantly to the paper. From Figure 10 there are two small clusters but there are many confounders that could contribute to this (e.g. antibiotic use or other treatments). To understand the relationship between samples and species other methods could be used such as interpretable machine learning (e.g. SHAP arXiv:1705.07874)**

While it is correct that microbiota clustering might be associated with confounders or some features of the community (e.g. enterotypes, modes of external drivers, etc.), we do not have an intention of an explainable feature discovery with this attempt. Indeed, clustering is conducted as an unbiased dimensionality reduction to regularize for the "large p, small n" classification problem.

A nonlinear dimension reduction technique, t-distributed Stochastic Neighbor Embedding (t-SNE) is a commonly used graphic approach to assist clustering methods such as k-means, with respect to determining the number of clusters and cluster memberships. In the revised manuscript, t-SNE is employed for visualizing the identified clusters (subgroups of IBD patients and healthy samples). To this end, based on the relative abundance values of the 14 identified species, two-dimensional t-SNE maps were generated separately for i) IBD patients, and ii) healthy samples (Figure 6). We performed visual inspection of cluster-colored tSNE plots. As shown in Figure 6 of the revised manuscript, the subgroups of IBD patients and the subgroups of healthy samples are distinct.

**Validity of the findings**

**• The flawed feature selection procedure used means all prediction estimates are invalid. They should be recalculated with the correct feature selection procedure, incorporating feature selection inside of the cross-validation resampling process**

Thanks for the point. In the revised version, we have splitted the data as training data and test data. We applied the feature selection algorithms on the training part. We tested our classification methods on the testing part. Actually, we have performed 100-fold Monte Carlo Cross Validation (MCCV) and the average of all performance metrics are calculated.

We have updated the manuscript and the results presented in Tables 1-3, Supplementary Tables 1-3 and all figures. We have also updated Figure 2 that illustrates the workflow of our method. In the revised Figure 2, we showed this splitting process.

**• Given the performance claims made about the models (AUROC approaching 1 in Figure 4) I think it is very important that the trained models are externally validated. There are many large metagenomic datasets available such as Gevers et al. 2014 which contains more than 1000 samples (10.1016/j.chom.2014.02.005). Previous machine learning work predicting IBD has used external validation to demonstrate the ability of trained models to generalise (10.1109/TCBB.2018.2831212). Qiita (https://qiita.ucsd.edu) is an excellent platform to browse and discover new microbiome datasets**

**When using external validation data it's important that the data can be harmonised across data sources (e.g. species or genus names match, and the same reference database is used)**

In addition to redesigning our experiments as requested by the reviewer and as explained in the previous question, we have evaluated the performance of our method using an external dataset. A multicenter gut metagenome dataset (Project accession: PRJEB1220) with samples collected from Denmark and Spain, containing Ulcerative colitis (UC) and Chrohn's disease (CD) cases as well as healthy controls were considered for validation. A random subset of samples of 50 UC patients, 50 CD patients (in total 100 IBD patients) and 100 healthy controls, containing more than 1 Gbp of sequencing reads were subject to taxonomic analysis. MetaPhlAn 3.0 is run using default parameters determining the relative abundances of all detected taxonomies. Very similar preprocessing protocol that is used for our exploration cohort (Accession: PRJEB2054), and that is presented in Methods section; is followed for the independent test cohort (Project accession: PRJEB1220). While using the validation data, we make sure that the species or genus names match, and the same reference database is used.

Among our 14 potential taxonomic biomarkers, 10 species (Supplementary Table 3) are found in the validation dataset. We compared the performance of these 10 features (species) against 10 random features using Random Forest classifier and 10 fold Monte Carlo Cross Validation. As shown in Table 1, on the validation data, the generated RF model resulted in higher performance metrics when our 10 selected features (species) are used, as compared to the randomly generated 10 features. Especially, in terms of specificity, on the validation data, using RF classifier, one can easily observe the sharp decrease to 0,49 when 10 random features are tested, as compared to the obtained specificity value of 0,86 with 10 selected species. The same trend of significant decrease in terms of specificity is observed in the results of all other tested classifiers (Table 1).

In our experiments, we also observed that 10 selected species performed slightly better in the Validation Cohort, compared to the Exploration Cohort. As presented in Table 2, those 10 selected species yielded higher performance metrics in Validation Cohort, compared to Exploration Cohort. Based on our results in Table 2, we can state that 4 additional features that are selected in the Exploration Cohort, but do not exist in the Validation Cohort contributed to the performance of the model. We have explained these findings in the results section of the revised manuscript.

## III. Responses to Reviewer #2 (Richard Gault):
**Basic reporting**

**• Further supporting information is required for the claims made in lines 38-40.**

We thank the reviewer for this suggestion. We have edited the sentences in the above-mentioned lines, removed our strong claims and added further supporting information.

**• There are some minor spelling and grammatical errors including:**

**Line 60: referencing style should be kept consistent throughout.**

**Line 297: Euclidean (E not e).**

**Line 308: Remove ?**

**Lines 325-326: Grammatical error "applying a careful feature selection". Please rephrase**

Thanks. We have corrected these sentences in the revised manuscript.

**• There are some difficulties with the following Figures:**

**Figure 4: Please use . instead of , for the decimal point. Please also add (%) as units to the header of the Accuracy columns. Increasing font size would be desirable here even if it means increasing the table sizes and significantly reducing the white space between them.**

**Figure 5: Colours are difficult to differentiate for those that are color blind. Please consider differentiating the bars with different fill textures/patterns. Please also add a label for the y-axis on each subfigure and please include confidence intervals.**

**Figure 6: The text in all three sub-figures is illegible. Note the x-axis label for (a) has been cut off by sub-figure (c). The numeric values in (c) are completely illegible and should be removed to aid clarity.**

**Figure 7: Please label y-axis. Please increase font size on x-axis. Please adjust the y-axis so that the numeric values are consistent on both.**

**Figure 8: Please label y-axis and increase the font size of the text on the right-hand side.**

**Figure 9: Please adjust the yellow coloring as it is difficult to see with the white background.**

**Figure 10: Perhaps this is best presented as a whole page image in landscape format?**

Based on the reviewer's comments, we have regenerated Figures 4-10 in the revised manuscript.

**Experimental design**

**• There are however some concerns over the scientific methodology.**

**The feature selection approach in its current form is at risk of leading to overfitting of the classification models. The feature selection process should be carried out as part of the resampling method. The methodology should be revised and the experimentation repeated once this change has been implemented.**

Thanks for the point. In the revised version, we have splitted the data as training data and test data. We applied the feature selection algorithms on the training part. We tested our classification methods on the testing part. Actually, we have performed 100-fold Monte Carlo Cross Validation (MCCV) and the average of all performance metrics are calculated. Accordingly, we have updated the manuscript and the results presented in Tables 1-2, Supplementary Tables 1-3 and redrawn the Figures. We have also updated Figure 2 that illustrates the workflow of our method. In the revised Figure 2, we showed this splitting process.

**• More information is required on the pre-processing of the raw MetaHit sequence data. Are there any further preprocessing steps conducted to refine the metagenomic data?**

We thank the reviewer for raising this point. Prior to taxonomic analysis with MetaPhlAn, we followed the standard quality analysis, which is proposed by the Human Microbiome Project (HMP)[4]. According to that, firstly the duplicate reads were removed by a modified version of EstimateLibraryComplexity in Picard tool package. The quality trimming was performed using TrimBWAStyle script[5], which is also recommended by the HMP SOP. We skipped the human genome contamination removal step, as the data provided is already free of human contamination. Prior to taxonomic analysis, we also filtered out the reads, which are smaller than 90bp in length.

We have considered the standard relative abundance normalization by dividing the read count of each taxonomic bin to the total number of the reads for a sample, so that taxonomic abundances are real numbers in the range of [0, 1], which sums up to 1 within each sample. Samples containing less than 1 million reads were discarded. We have added this explanation to the revised manuscript.

**• How were patients classed as having (or not having IBD)? How homogenous was the IBD patient cohort (and indeed the control cohort)?**

In the original study that we obtained the metagenomics data (PRJEB2054), it is stated that: "Healthy controls were recruited among family relatives of IBD patients; antibiotic treatment for at least 4 weeks before fecal sample collection was excluded. IBD subjects were in clinical remission for at least 3 months, and had stable maintenance therapy with mesalazine or azathioprine."

The patient demographics providing the country, gender, age, body-mass index are available in the original study. In our revised manuscript, we added such a sentence.

 **Validity of the findings**

**• The primary concern here follows from the risk of overfitting caused by the methodological approach (as described in my comment above). The results should be revisited once the methodology is revised. There are some general comments which would be beneficial for the authors to consider in their revision.**

Thanks for your suggestion. As we answered above, in the revised version, we have splitted the data as training data and test data. We applied the feature selection algorithms on the training part. We tested our classification methods on the testing part. Actually, we have performed 100-fold Monte Carlo Cross Validation (MCCV) and the average of all performance metrics are calculated. Based on our revision of the method, we have updated the manuscript and the results presented in Tables 1-2, Supplementary Tables 1-3 and regenerated the figures. We have also updated Figure 2 that illustrates the workflow of our method. In the revised Figure 2, we showed this splitting process.
* * *
[4] Human Microbiome Project Consortium. Structure, function and diversity of the healthy human microbiome. Nature 2012;486(7402):207–214. PMID: 22699609
https://www.hmpdacc.org/hmp/doc/ReadProcessing_SOP.pdf
[5] https://github.com/dgpinheiro/bioinfoutilities/blob/master/TrimBWAStyle.pl

**• Whilst the sample size is reasonable for this study, any findings from this work (with the revised methodology) would be significantly strengthened by validating the findings on a larger open-sourced dataset. Whilst this is substantial work it will result in more robust findings, support model comparisons in the future, and ease reproducibility, which will hopefully see the author's efforts rewarded through increased community adoption of their approach.**

In addition to redesigning our experiments as requested by the reviewer and as explained in the previous question, we have evaluated the performance of our method using an external dataset. A multicenter gut metagenome dataset (Project accession: PRJEB1220) with samples collected from Denmark and Spain, containing Ulcerative colitis (UC) and Chrohn's disease (CD) cases as well as healthy controls were considered for validation. A random subset of samples of 50 UC patients, 50 CD patients (in total 100 IBD patients) and 100 healthy controls, containing more than 1 Gbp of sequencing reads were subject to taxonomic analysis. MetaPhlAn 3.0 [1] is run using default parameters determining the relative abundances of all detected taxonomies. Very similar preprocessing protocol that is used for our exploration cohort (Accession: PRJEB2054), and that is presented in Methods section; is followed for the independent test cohort (Project accession: PRJEB1220). While using the validation data, we make sure that the species or genus names match, and the same reference database is used.

Among our 14 potential taxonomic biomarkers, 10 species (Supplementary Table 3) are found in the validation dataset. We compared the performance of these 10 features (species) against 10 random features using Random Forest classifier and 10 fold Monte Carlo Cross Validation. As shown in Table 1, on the validation data, the generated RF model resulted in higher performance metrics when our 10 selected features (species) are used, as compared to the randomly generated 10 features. Especially, in terms of specificity, on the validation data, using RF classifier, one can easily observe the sharp decrease to 0,49 when 10 random features are tested, as compared to the obtained specificity value of 0,86 with 10 selected species. The same trend of significant decrease in terms of specificity is observed in the results of all other tested classifiers (Table 1).

In our experiments, we also observed that 10 selected species performed slightly better in the Validation Cohort, compared to the Exploration Cohort. As presented in Table 2, those 10 selected species yielded higher performance metrics in Validation Cohort, compared to Exploration Cohort. Based on our results in Table 2, we can state that 4 additional features that are selected in the Exploration Cohort, but do not exist in the Validation Cohort contributed to the performance of the model. We have explained these findings in the results section of the revised manuscript.

**Line 250-251: Please comment on the biological relevance of the features contained in the intersection of all feature selection methods. Is XGBoost good because it identifies particularly interesting species or are the really interesting species contained within the 10 species of the intersection?**

Based on the reviewer's suggestion, in the discussions section of the revised manuscript, we have commented on the biological relevance of the features contained in the intersection of feature selection methods. As presented in Figure 4 and Supplementary Table 1 of the revised manuscript, the performance of XGBoost, Select K Best and Information Gain is good in terms of minimizing the microbiota. Interesting species are contained within the 14 species of the

intersection. In the discussions section of the revised manuscript, we compared in detail our findings (14 candidate taxonomic biomarkers of IBD) with literature. The relevance of 14 selected species in terms of IBD development is discussed in detail in the revised manuscript.

**Line 270: Please elaborate on how the values 12 and 7 were identified. They are not immediately obvious from Figure 7.**

Based on the reviewer's comments, we have redesigned our feature selection process and incorporated 100-fold Monte Carlo Cross Validation (MCCV). Accordingly, in the revised manuscript, we have detected 14 informative species and we have repeated our k-means clustering analysis using those 14 features.

In our analysis, we have utilized the Euclidean distance metric and Elbow method[6] in order to determine the optimum number of clusters. In this method, the point where the decline in the error slows down indicates the optimum number of clusters. As shown in Supplementary Figure 3 of the revised manuscript, four subgroups among controls and three subgroups among IBD patients were discovered. A nonlinear dimension reduction technique, t-distributed Stochastic Neighbor Embedding (t-SNE) is a commonly used graphic approach to assist clustering methods such as k-means, with respect to determining the number of clusters and cluster memberships. In the revised manuscript, t-SNE is employed for visualizing the identified clusters (subgroups of IBD patients and healthy samples). To this end, based on the relative abundance values of the 14 identified species, two-dimensional t-SNE maps were generated separately for i) IBD patients, and ii) healthy samples (Figure 6). We performed visual inspection of cluster-colored tSNE plots. As shown in Figure 6, the subgroups of IBD patients and the subgroups of healthy samples are distinct.

**Lines 263-281: Please comment on the role the isolated species are known to have in biology similar to the discussion presented in lines 258-262.**

Thanks for the point. In the revised manuscript, we have discussed the potential roles of the identified species. In the discussions section of the revised manuscript, we compared in detail our findings (candidate taxonomic biomarkers of IBD) with literature. The relevance of 14 selected species in terms of IBD development is discussed in detail in the revised manuscript.

**Line 298: Why cut off at the top 24 species? This seems arbitrary without further explanation.**

Based on the reviewer's comments, we have updated the methodology and the results presented in Tables 1-2, Supplementary Tables 1-3 and regenerated all figures. We have also updated Figure 2 that illustrates the workflow of our method. In the revised manuscript, we have utilized six different feature selection methods, i.e., Conditional Mutual Information Maximization (CMIM), Fast Correlation Based Filter (FCBF), Min Redundancy Max Relevance (mRMR), Select K Best (SKB), Information Gain (IG) and Extreme Gradient Boosting (XGBoost). For each feature selection method, we focused on the top 100 features. In order to evaluate the effects of different classification methods, we have used Decision Tree, Random Forest, LogitBoost, AdaBoost, an ensemble of SVM with kNN, and an ensemble of the Logitboost with
* * *
[6] https://predictivehacks.com/k-means-elbow-method-code-for-python/

kNN. We optimize the parameters "c" and "gamma" for SVM, "number of tree" for Decision tree, Random Forest, Logitboost and Adaboost. By using several metrics as described in Methods section, we have compared the performances of different classifiers using (i) all features (without feature selection); (ii) top 100 features selected using CMIM, mRMR, FCBF, SKB, IG and XGBoost (presented in Supplementary Table 1). As shown in the same table, in our experiments with IBD-associated metagenomics data, SKB, IG and XGBoost feature selection methods resulted in high accuracy, F1 measure, sensitivity, recall values, and high AUC scores for different classifiers. To find the most relevant and informative features, for each feature, we calculated the scaled importance values for three selected feature selection methods (SKB, IG and XGBoost) across different classifiers (presented in Supplementary Table 2). To eliminate the lowest ranking features among the top 100, we applied scaled importance value cutoff of 0.5. As shown in Figure 3, we ended up with 23, 57, 96 selected features in SKB, IG and XGBoost feature selection methods, respectively. 14 of those features were commonly identified in all three feature selection methods.

As shown in Figure 4 and Supplementary Table 1 of the revised manuscript, compared to other classifiers, RF classifier generated higher performance results for those promising feature selection methods (SKB, IG and XGBoost) and for 14 selected features. Since the tree model is easy for interpretation and since one can easily convert the model into a rule set, in our further experiments, we decided to continue with the RF classifier in our further experiments. Additionally, RF is one of the most used algorithms in the human microbiome studies as reported by Marcos-Zambrano et al. (2021). As shown in Figure 5, the generated RF model resulted in 0.85 F1-score, 0.92 AUC, and 87% accuracy when all 1331 features are used (without applying feature selection methods). By only using the 14 features that are commonly selected in three promising feature selection methods, 0.85 F1-score, 0.93 AUC, and 88% accuracy metrics were obtained. Compared to using all features, those selected 14 features performed 1% higher in terms of accuracy and AUC metrics; 5% higher in terms of specificity and precision metrics; as shown in Figure 5 and Supplementary Table 1. The model using only those 14 species resulted in the same F1-score (0.85) with the F1-score obtained using all features. In other words, IBD diagnosis could be possible with 88% accuracy by checking only the amounts of 14 specific species among 1331 different species. Checking the amounts of fewer features means less time and cost. Hence, we proposed those 14 features (species) shown in Figure 3 as potential taxonomic biomarkers for IBD.

**Validation on external data**

We have evaluated the performance of our method using an external dataset. A multicenter gut metagenome dataset (Project accession: PRJEB1220) with samples collected from Denmark and Spain, containing Ulcerative colitis (UC) and Chrohn's disease (CD) cases as well as healthy controls were considered for validation. A random subset of samples of 50 UC patients, 50 CD patients (in total 100 IBD patients) and 100 healthy controls, containing more than 1 Gbp of sequencing reads were subject to taxonomic analysis. MetaPhlAn 3.0 is run using default parameters determining the relative abundances of all detected taxonomies. Very similar preprocessing protocol that is used for our Exploration Cohort (Accession: PRJEB2054), and that is presented in Methods section; is followed for the independent test data (Project accession: PRJEB1220). While using the validation data, we make sure that the species or genus names match, and the same reference database is used. Among our 14 potential taxonomic biomarkers,

10 species (Supplementary Table 3) are found in the validation dataset. We compared the performance of these 10 features (species) against 10 random features using Random Forest classifier and 10 fold Monte Carlo Cross Validation. As shown in Table 1, on the validation data, the generated RF model resulted in higher performance metrics when our 10 selected features (species) are used, as compared to the randomly generated 10 features. Especially, in terms of specificity, on the validation data, using RF classifier, one can easily observe the sharp decrease to 0,49 when 10 random features are tested, as compared to the obtained specificity value of 0,86 with 10 selected species. The same trend of significant decrease in terms of specificity is observed in the results of all other tested classifiers (Table 1).

In our experiments, we also observed that 10 selected species performed slightly better in the Validation Cohort, compared to the Exploration Cohort. As presented in Table 2, those 10 selected species yielded higher performance metrics in Validation Cohort, compared to Exploration Cohort. Based on our results in Table 2 we can state that 4 additional features that are selected in the Exploration Cohort, but do not exist in the Validation Cohort contributed to the performance of the model.

These findings are explained in detail in the results section of the revised manuscript.

**Discussion Section:**

**Lines 325-326: The description of "careful feature selection" is perhaps misleading. The authors apply a number of feature selection methods, but what is the take-home message for the reader? Which feature selection method should be used? Is it important to select a particular feature selection method depending on the classification model to be used? Should a number of feature selection methods be used and the final features are determined by the union/intersection?**

Feature selection has proven to be a successful preprocessing tool for machine learning problems. However, choosing between the growing number of selection methods is challenging. Different feature selection methods have their advantages/disadvantages as discussed in different reviews. In a previous work (Bolon-Canedo et. al., 2013), several state-of-the-art feature selection methods were reviewed in terms of their ability to solve common problems such as correlation and redundancy, data nonlinearity, noise in the input features, noise in the target class, and having a number of features much higher than the number of samples. Feature selection methods have different usages in the biomedical domain, as presented in (Remeseiro and Bolon-Canedo, 2019), (Manikandan and Abirami, 2021).

In metagenomics studies, the number of predictors (number of taxa) is much more than the number of observations (samples). In this respect, some metagenomics studies focus on the feature selection process rather than the classification. Although feature selection for meta-genome-based disease prediction seems to be a less-explored area, it may be just as important as the classification method used and may enhance interpretability, motivating further research in this direction. For metagenomics studies, there is no consensus on which feature selection method should be used as explained in a recent review paper (Marcos-Zambrano et. al., 2021). We think that a number of feature selection methods should be used and the final features are determined by the intersection. We have added this explanation to our revised manuscript.

Also please see our detailed answer to the previous question. In summary, in the revised manuscript, we have utilized six different feature selection methods, i.e., Conditional Mutual Information Maximization (CMIM), Fast Correlation Based Filter (FCBF), Min Redundancy Max Relevance (mRMR), Select K Best (SKB), Information Gain (IG) and Extreme Gradient Boosting (XGBoost). By using several metrics, we have compared the performances of different classifiers using (i) all features (without feature selection); (ii) top 100 features selected using CMIM, mRMR, FCBF, SKB, IG and XGBoost (presented in Supplementary Table 1). As shown in the same table, in our experiments with IBD-associated metagenomics data, SKB, IG and XGBoost feature selection methods resulted in high accuracy, F1 measure, sensitivity, recall values, and high AUC scores for different classifiers. As shown in Figures 4 and 5, the generated RF model resulted in 0.85 F1-score, 0.92 AUC, and 87% accuracy when all 1331 features are used (without applying feature selection methods). By only using the 14 features that are commonly selected in three promising feature selection methods, 0.85 F1-score, 0.93 AUC, and 88% accuracy metrics were obtained. Checking the amounts of fewer features means less time and cost. Hence, we proposed those 14 features (species) shown in Figure 3 as potential taxonomic biomarkers for IBD. Additionally, we have evaluated the performance of our method using an external dataset (as explained in detail in our answer to the previous question).

**Lines 330 and 334: "unprecedented performance/results" - This is a bold statement and should be revised. The conclusion "feature selection can improve classification results" is a well-established finding so there is precedence that your results should be higher than previous studies. The performance increase is certainly good but I don't think unprecedented is the appropriate description.**

Thanks for the point. In the revised manuscript we removed the "unprecedented performance/results" statement.

**Line 339: "leads to a framework" - perhaps the rephrase along the lines of "provides the foundation for a framework to be developed in future work" would be more appropriate.**

Based on the reviewer's suggestion, in the revised manuscript we have paraphrased this statement.

**Line 341: "would imply a potential for narrowing" - I don't think it is clear what is meant here. I think a rephrase would be helpful for the reader.**

Based on the reviewer's suggestion, in the revised manuscript we have removed the above-mentioned unclear sentence.

**Line 352: "provides a blueprint". I think one of the following may be more appropriate: "provides evidence" or "provides justification" or "provides motivation" or "provides support".**

Based on the reviewer's suggestion, in the revised manuscript we have paraphrased this statement.

**Comments for the Author**

**The paper is well written and I would compliment the authors for some good practices, such as Figures 2 and 3 which clearly and concisely illustrate the concept outlined in the main discussion. The paper presents promising results however, further experimentation should be conducted to ensure the classification models are not inadvertently overfitted as a result of the feature selection approach outlined in the current manuscript.**

We appreciate the reviewer's positive feedback. Still, in order to ensure the classification models are not inadvertently overfitted, in the revised version, we have splitted the data as training data and test data. We applied the feature selection algorithms on the training part. We tested our classification methods on the testing part. Actually, we have performed 100-fold Monte Carlo Cross Validation (MCCV) and the average of all performance metrics are calculated. Based on our revision of the method, we have updated the manuscript and the results presented in Tables 1-2, Supplementary Tables 1-3 and regenerated all figures. We have also updated Figure 2 that illustrates the workflow of our method. In the revised Figure 2, we showed this splitting process.

## IV. Responses to Reviewer #3 (P Morell Miranda):

### Basic reporting

**The reporting of this article is, to describe it briefly, unbalanced. It is written in clear, professional English, and there are sections of well developed and explained text, both in Materials and Methods and in the Experiments (according to PeerJ standards it should be named Results), but there are other sections that are clearly missing content, or the one written is contradicting itself. As an example, we do not need to look further than the Abstract, where we can find a claim that dysbiosis in the cause of IBD and other gastrointestinal illnesses, to a few lines later read that we do not have a clear picture of how this dysbiosis affects IBD, just that they show a strong association (this is also repeated in the Introduction). Claims of causality should be avoided unless evidence of this relationship is abundant, and with the gut metagenome we are just not there yet.**

Based on the reviewer's comments we have renamed the "Experiments" section with "Results". In the revised version, we have rewritten several sections. We broaden the scope of the introduction section.

**Getting into the Introduction, I find that some general description of what we know about the relationships between IBD and dysbiosis gut environments would be beneficial to highlight the direct need for this kind of diagnosis approaches. I would add a brief description of the most common symptoms of IBD and the relationship between this illness and the gut to lines 51 to 57. The mention of the high comorbidity of IBD with other dysbiosis related illnesses, such as depression, anxiety or obesity could also highlight the relevance of this research.**

Thanks for your suggestions. Based on your suggestions, in the revised version, we have included detailed paragraphs to the Introduction section. We have added a general description of what we know about the relationships between IBD and dysbiosis gut environments; we have highlighted the direct need for this kind of diagnosis approaches. We have added a brief

description of the most common symptoms of IBD; the relationship of IBD and the gut; the high comorbidity of IBD with other dysbiosis related illnesses, such as depression, anxiety or obesity.

**Regarding the figures, I do not have any major complaints. The tables in Figure 4 may be too small in some screens, so maybe they should be independent supplementary tables. Other than that, they are clear, well described and easy to interpret.**

We thank the reviewer for his/her kind words. We have redrawn Figure 4 based on our revised experiments. In the revised version, we also provide the details of these results as independent supplementary tables.

**Raw data was supplied through the project's website. This should still be disclosed on the paper.**

Based on the reviewer's suggestion, in the revised version, we disclose that the raw data was supplied through the project's website.

Experimental Design

**The experimental design follows the trend of using different Machine Learning algorithms for classification of different atypical markers of illnesses. This is a powerful approach both to understand better which alterations in the microbiotic environment are most influential on IBD, and for clinical purposes as diagnostic and treatment.**

**The research question is well defined and relevant, and the investigation is, as far as I can see, performed according to technical and ethical standards.**

We thank the reviewer for his/her kind words.

**In the description of the methods we find, again, this unbalance that we previously saw in the introduction and abstract. The first part of that section (lines 122-144) and the Feature Selection descriptions are good. Model Selection and Unsupervised Learning, however, are too brief and shallow. There is no description of what any of the classifiers used, nor a justification of their use. In fact, the Model Construction section is composed mostly of a description of the performance metrics used to analyze how well those algorithms performed, instead of why those algorithms were used and how they work. Something similar happens with the Unsupervised learning, where the description is so brief that a description of the algorithms has been included in the Experiments (Results).**

In the revised manuscript, we have extended the Model Construction and Unsupervised Learning Subsections of Methodology Section. Applying supervised learning to the human gut microbiome can help us to detect subsets of microorganisms that are highly discriminative. Accordingly, one can build prediction models that can accurately classify unlabeled samples. Recently, (Marcos-Zambrano et al., 2021) reviewed machine learning (ML) applications for microbiome studies via analyzing 89 papers. They reported that the most common supervised learning algorithms that were used for microbiome analysis were Random Forest (RF), Support Vector Machines (SVM), Logistic Regression (LR) and k-NN (k nearest neighbor). They concluded that there are several factors that need to be considered during the selection of the ML

algorithm (i.e., number of features, number of observations, data quality, data type etc.), and they recommend applying and evaluating more than one method and selecting the one with the best performance. Using 29 human microbiome benchmark datasets, another recent study by (Wang et al, 2020) compared the performances of two ensemble methods (Random Forests (RF), eXtreme Gradient Boosting decision trees (XGBoost)), and two traditional methods (The elastic net (ENET) and Support Vector Machine (SVM)). They find that XGBoost outperforms all other methods in a few benchmark datasets; and XGBoost, RF and ENET displayed comparable performance in the remaining benchmark datasets. Along this line, to discriminate IBD samples from controls, we construct a range of machine learning models using different classification algorithms (Random Forest (RF), Decision Tree, Logitboost, Adaboost, Support Vector Machine (SVM) and stacking ensemble classifiers (i.e., an ensemble of SVM with kNN (k nearest neighbor), an ensemble of the Logitboost with kNN)). The working principles of these classification algorithms are also summarized in the methodology section of the revised manuscript.

**Validity of the Findings**

**To my knowledge the results presented in this paper are novel and relevant. I would like the authors to take a bit of time to try to link them to other research in IBD and metagenomics, mostly on the topics I already mentioned on my comments on Basic Reporting. IBD is a complex, multifactorial and comorbid illness, and that should be addressed both in the introduction and description of the issue, and on the discussion and interpretation of our results, even if the scope of our analysis does not include all those factors or comorbid conditions.**

Thanks for your suggestions. In the revised version, we have included these descriptive statements to the both Introduction and Discussion sections.

**I must admit that I do not like to see some of the results, as the most relevant species for IBD diagnosis according to your analysis, firstly mentioned in the Discussion. Maybe restructuring the article according to the PeerJ Standard Sections template would help avoid this. I understand that the focus of this paper is on the use of Machine Learning as a diagnostic tool, but if you are going to mention it at all, it still should be mentioned in the Results section.**

Thanks for the point. As suggested by the reviewer, we have restructured our revised manuscript according to the PeerJ Standard Sections template. In the revised manuscript, firstly we have presented our findings such as the most relevant species for IBD diagnosis in the "Results" section, before more thoroughly discussing them in the "Discussions" section. In the discussions section of the revised manuscript, we compared in detail our findings (candidate taxonomic biomarkers of IBD) with literature. The relevance of 14 selected species in terms of IBD development is discussed in detail in the revised manuscript.

**References:**

Bolón-Canedo, V., Sánchez-Maroño, N., Alonso-Betanzos, A. (2013). A review of feature selection methods on synthetic data, Knowl. Inf. Syst., 34 (3), pp. 483-519.

Erb, I., and Notredame, C. (2016). How should we measure proportionality on relative gene expression data? Theory Biosci. 135, 21–36. doi: 10.1007/s12064-015-0220-8

Friedman, J., and Alm, E. J. (2012). Inferring correlation networks from genomic survey data. PLoS Comput. Biol. 8:e1002687. doi: 10.1371/journal.pcbi.1002687.

Gloor GB, Macklaim JM, Pawlowsky-Glahn V and Egozcue JJ (2017) Microbiome Datasets Are Compositional: And This Is Not Optional. Front. Microbiol. 8:2224. doi: 10.3389/fmicb.2017.02224.

Lovell, D., Pawlowsky-Glahn, V., Egozcue, J. J., Marguerat, S., and Bähler, J. (2015). Proportionality: a valid alternative to correlation for relative data. PLoS Comput. Biol. 11:e1004075. doi: 10.1371/journal.pcbi.1004075.

Manikandan, G., and Abirami, S. (2021) Feature Selection and Machine Learning Models for High-Dimensional Data: State-of-the-Art. Computational Intelligence and Healthcare Informatics, 43.

Marcos-Zambrano, L. J., Karaduzovic-Hadziabdic, K., Przymus, P., Trajkovik, V., Aasmets, O., Berland, M., et al. (2021). Applications of machine learning in human microbiome studies: a review on feature selection, biomarker identification, disease prediction and treatment. Front. Microbiol. 12:634511. doi: 10.3389/fmicb.2021.634511.

Remeseiro, B., and Bolon-Canedo, V. (2019). A review of feature selection methods in medical applications, Computers in Biology and Medicine, Volume 112, 103375, https://doi.org/10.1016/j.compbiomed.2019.103375.

Wanga, X.W. and Liu, Y.Y. (2020) Comparative study of classifiers for human microbiome data, Medicine in Microecology, 4, 100013.

---

## Round 0.3 · Minor Revisions

Please note that language problems still remain in your manuscript. For example, "MetaPhlAn2 tool (Ditzler et al., 2015b) is widely in literature" (lines 189-190) [should be widely USED]; "...Decision Tree, Random Forest, LogitBoost, AdaBoost, an ensemble of SVM with kNN, and an ensemble of the Logitboost with kNN is considered" (lines 404-405) [should be ARE considered]. These are just two examples; thorough checking of the entire text should be carried out, preferably by a fluent English speaker.

---

## Round 0.4 · accepted · Accept

The language revisions made were satisfactory.